# Past terrestrial hydroclimate sensitivity controlled by Earth system feedbacks

Ran Feng [1✉], Tripti Bhattacharya [2], Bette L. Otto-Bliesner [3], Esther C. Brady[3], Alan M. Haywood[4], Julia C. Tindall [4], Stephen J. Hunter [4], Ayako Abe-Ouchi [5], Wing-Le Chan [5], Masa Kageyama[6], Camille Contoux[6], Chuncheng Guo[7], Xiangyu Li[8], Gerrit Lohmann [9,10], Christian Stepanek [9], Ning Tan[11], Qiong Zhang [12], Zhongshi Zhang[8], Zixuan Han[13], Charles J. R. Williams [14], Daniel J. Lunt [14], Harry J. Dowsett [15], Deepak Chandan [16] & W. Richard Peltier[16]

Despite tectonic conditions and atmospheric $CO_2$ levels ($pCO_2$) similar to those of present-day, geological reconstructions from the mid-Pliocene (3.3-3.0 Ma) document high lake levels in the Sahel and mesic conditions in subtropical Eurasia, suggesting drastic reorganizations of subtropical terrestrial hydroclimate during this interval. Here, using a compilation of proxy data and multi-model paleoclimate simulations, we show that the mid-Pliocene hydroclimate state is not driven by direct $CO_2$ radiative forcing but by a loss of northern high-latitude ice sheets and continental greening. These ice sheet and vegetation changes are long-term Earth system feedbacks to elevated $pCO_2$. Further, the moist conditions in the Sahel and subtropical Eurasia during the mid-Pliocene are a product of enhanced tropospheric humidity and a stationary wave response to the surface warming pattern, which varies strongly with land cover changes. These findings highlight the potential for amplified terrestrial hydroclimate responses over long timescales to a sustained $CO_2$ forcing.

[1] Department of Geosciences, College of Liberal Arts and Sciences, University of Connecticut, Mansfield, CT, USA. [2] Department of Earth and Environmental Sciences, Syracuse University, Syracuse, NY, USA. [3] Climate and Global Dynamics Laboratory, National Center for Atmospheric Research, Boulder, CO, USA. [4] School of Earth and Environment, University of Leeds, Woodhouse Lane, Leeds, West Yorkshire LS29JT, UK. [5] Atmosphere and Ocean Research Institute, University of Tokyo, Kashiwa, Japan. [6] Laboratoire des Sciences du Climat et de l'Environnement, LSCE/IPSL, CEA-CNRS-UVSQ, Université Paris-Saclay, F-91191 Gif-sur-Yvette, France. [7] NORCE Norwegian Research Centre, Bjerknes Centre for Climate Research, 5007 Bergen, Norway. [8] Department of Atmospheric Science, School of Environnemental Studies, China University of Geoscience, Wuhan 430074, China. [9] Alfred Wegener Institute-Helmholtz Centre for Polar and Marine Research, Bremerhaven, Germany. [10] University of Bremen, Bremen, Germany. [11] Key Laboratory of Cenozoic Geology and Environment, Institute of Geology and Geophysics, Chinese Academy of Sciences, Beijing 100029, China. [12] Department of Physical Geography and Bolin Centre for Climate Research, Stockholm University, Stockholm, Sweden. [13] College of Oceanography, Hohai University, Nanjing, China. [14] School of Geographical Sciences and Cabot Institute, University of Bristol, University Road, Bristol BS8 1SS, UK. [15] Florence Bascom Geoscience Center, U. S. Geological Survey, Reston, VA, USA. [16] Department of Physics, University of Toronto, Toronto, CA, Canada. ✉email: ran.feng@uconn.edu

Geologic evidence suggests dramatic reorganizations of subtropical climate during past greenhouse climate intervals, including the mid-Piacenzian Warm Period (3.3–3.0 Ma, commonly referred to as the mid-Pliocene). Multiple proxies of hydroclimate indicate large, deep lakes and reduced dust flux across North Africa during the mid-Pliocene[1,2], and more mesic vegetation in South and East Asia[3,4]. Additional sedimentary and paleobotanical data also points to wetter subtropical Eurasia conditions prior to the intensification of northern hemisphere glaciation (Supplementary Table 1). These changes imply large increases in precipitation (P) minus evaporation (E), referred to as P–E. All are associated with $p$CO$_2$ levels of approximately 400 ppm[5,6], similar to today's level. Elevated $p$CO$_2$ can lead to moderate increases in precipitation across these regions as a result of tropospheric moistening[7], enhanced land–sea thermal contrast[8], and enhanced inland moisture advection[9,10]. However, evaporation also increases due to surface warming. As a result, predicted changes in terrestrial water balance (P–E), and associated changes in soil moisture and runoff remain equivocal across subtropical continents[11].

Modeled hydroclimate responses to CO$_2$ increases broadly follow the "wet-gets-wetter, dry-gets-drier" paradigm[7] with strong modulations from land warming patterns, the land-sea warming contrast[12], tropical SST patterns[13], and feedbacks from soil moisture and CO$_2$ fertilization effects on leaf phenology[14,15]. In simulations featuring middle-of-the-road future warming scenarios, predicted changes in P–E by the end of the 21st century (2081–2100) are minimal across the subtropical Sahel and East Asia relative to both the historical period (1986–2005) (Fig. 1a and d) and preindustrial (PI) (Supplementary Fig. 1). These future scenarios are often thought to be comparable to the mid-Pliocene climate[16,17](Fig. 1a). Sources for the disparity between the hydroclimate state recorded by mid-Pliocene proxies and future simulations are unknown. One potential source might be the transient nature of future climate change as opposed to the quasi-equilibrium nature of mid-Pliocene climate. Yet, even equilibrium simulations with only the mid-Pliocene CO$_2$ forcing fail to produce moist subtropical terrestrial conditions[18]. Our simulation also confirms this result (Fig. 1).

From the perspective of atmospheric dynamics, two leading hypotheses have been proposed to explain the wetter subtropical continents during past warm climates. Both hypotheses were proposed when many older-generation models at lower spatial resolutions were not able to simulate wetter subtropical continents. One hypothesis emphasizes the hydroclimate impact of an El Niño-like Pacific mean state. SST records from the tropical Pacific document greater warming across the eastern equatorial Pacific than the western Pacific warm pool[19–23], resulting in an El-Niño-like pattern of SST anomalies. An El Niño-like Pacific SST pattern may strengthen and shift the subtropical jet equatorward, enhancing the transient eddy-driven moisture convergence and ascent[24,25]. The other hypothesis focuses on the role of a sluggish Hadley Circulation that results from a relaxed meridional SST gradient[26–28]. A weaker Hadley Circulation may lead to weakened zonal mean moisture divergence from the subtropics and, in turn, reduced aridity[18,29,30]. Nonetheless, with advancements in model development and boundary conditions, newer earth system model simulations show substantial differences in simulated past climate states[23,31,32], particularly in mid-Pliocene SST patterns[33], polar warmth[34], Atlantic Overturning Circulation[35], and precipitation[36].

Here, using atmosphere-ocean coupled global climate model (GCM) simulations from the most recent Pliocene Model Intercomparison Project Phase II (PlioMIP2)[33,37,38], we demonstrate that most current generation GCMs can reproduce the pattern of mid-Pliocene hydroclimate of the subtropical Sahel and East Asia

suggested by proxies without any paleoclimate-specific changes to model parameterizations. We focus on these regions given the large, coherent signal across the majority of PlioMIP2 simulations and the convergence between proxy data and model simulations (Fig. 1b, c). We further develop several new simulations using the Community Earth System Model version 2[35,39] to explore the extent to which simulated mid-Pliocene hydroclimate changes can be generated by changes in CO$_2$, tectonics, or vegetation and ice sheets. In contrast to previous hypotheses, wetter mid-Pliocene conditions in the subtropical Sahel and East Asia are driven by tropospheric moistening and changes to stationary wave dynamics in response to surface warming patterns due to vegetation and ice sheet changes. Both changes are part of the long-term Earth system feedbacks to a sustained CO$_2$ forcing. Consequently, model skill at simulating Pliocene hydroclimate states strongly scales with earth system sensitivities (ESS) of individual models instead of the equilibrium climate sensitivities (ECS).

## Results

**δ(P–E) pattern in models and proxies.** The last 100 years of simulations by 13 PlioMIP2 GCMs (Supplementary Table 2) were averaged to produce the ensemble mean. Mid-Pliocene changes in P–E, referred to as δ(P–E), and other climate variables are calculated with respect to preindustrial (PI) values. A robust moistening signal that is larger than the intermodel variability is found across the Sahel and subtropical Eurasia (Fig. 1b). This pattern is most pronounced during the boreal summer months (June to September) (Fig. 1c), with little or opposite changes during the boreal winter months (December to March) (Fig. S2). The spatial continuity of positive δ(P–E) from North Africa to subtropical East Asia is not a visual coincidence: models that show a large precipitation increase in the Sahel also tend to show a large precipitation increase in subtropical East Asia (Fig. 1d), indicating similar processes driving hydroclimate changes in both regions. We confirm this with moisture budget diagnostics (see below).

To compare modeled patterns of hydroclimate change to available geologic data, we compiled proxy indicators of mid-Pliocene terrestrial hydroclimate, drawing on existing compilations[18,40,41] as well as our own literature search. We identify a total of 64 proxy records that include sedimentological indicators, palynological, floral or faunal, offshore marine records, and stable isotope analyses of organic and inorganic materials (Table S1 and Fig. S3). These records are interpreted as qualitative or semi-quantitative indicators of mean hydroclimate state given the lack of orbital-scale variabilities documented in these records (see Methods). We quantify the extent to which proxies and models produce the same patterns of wetter, drier, or unchanged mid-Pliocene hydroclimate using a metric known as Gwet's AC designed for categorical data (see Methods). To account for the unknown sensitivity of proxy hydroclimate indicators to Pliocene P–E changes, model values of δ(P–E) are expressed as % changes from PI values at the proxy sites.

The annual pattern of multi-model mean mid-Pliocene δ(P–E) shows statistically significant agreement with proxy indicators of hydroclimate across the subtropical Sahel and East Asia for a wide range of δ(P–E) thresholds (Fig. 2a). This agreement is strongly driven by the simulated summer δ(P–E) pattern (Fig. 2c). Additionally, a subset of experiments featuring dynamic phenology and terrestrial carbon cycle also show a strong coupling between the positive δ(P–E) and enhanced net-primarily productivity across both regions, consistent with paleoecological reconstructions (Fig. S5). Yet, different models show varying skills at capturing proxy hydroclimate changes (Fig. 2a). For instance,

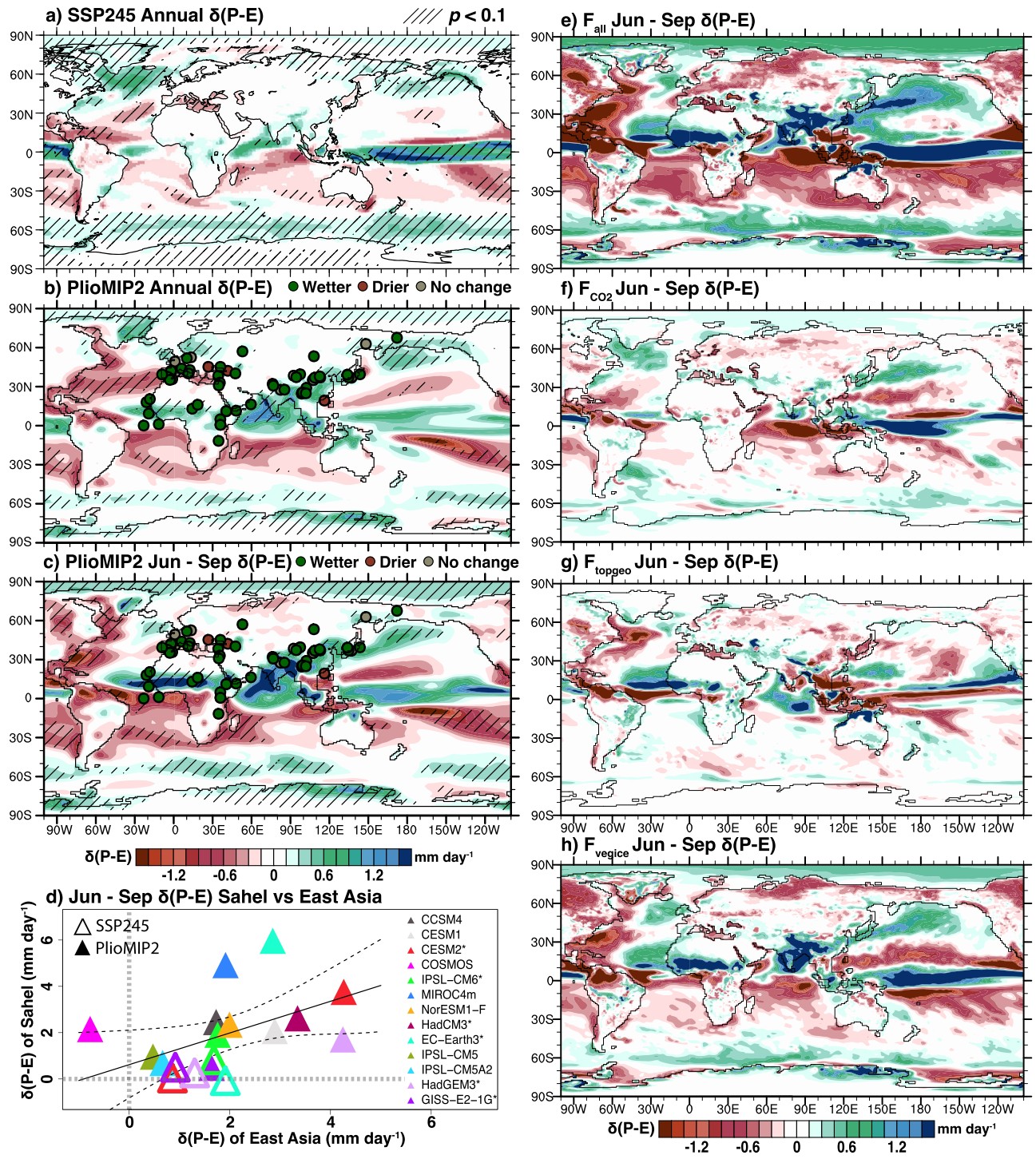

**Fig. 1 Hydroclimate changes in mid-Pliocene proxy records and simulations of mid-Pliocene and future climate. a** Changes in precipitation minus evaporation ($\delta(P-E)$, mm day$^{-1}$) between 2081 to 2100 and 1986 to 2005 following Shared Socioeconomical Pathway (SSP) 2–4.5 simulated by models participating in Climate Model Intercomparison Project 6. **b, c** Mid-Pliocene annual and boreal summer (June to September) mean $\delta(P-E)$ of the PlioMIP2 ensemble relative to the preindustrial simulated by the same models. Proxy data displayed as filled circles. **d** Correspondence between the subtropical Sahel (10°–20°N, 10°W–25°E) and East Asia (20°–30°N, 80°E–100°E) $\delta(P-E)$ simulated by PlioMIP2 experiments and a subset of SSP2–4.5 experiments with the same models (model names are marked with asterisks). **e–h** $\delta(P-E)$ in response to full Pliocene climate forcing conditions ($F_{all}$), $CO_2$ forcing alone ($F_{CO2}$), changes in geography and topography ($F_{geotop}$), and changes in vegetation and icesheet ($F_{vegice}$) simulated by Community Earth System Model version 2. The SSP2–4.5 ensemble includes models of BCC-CSM2-MR, CESM2, CESM2-WACCM, CanESM5, CNRM-CM6-1, CNRM-ESM2-1, EC-Earth3.3, GISS-E2-1-G, HadGEM3-GC31-LL, IPSL-CM6A-LR, MIROC6, MIROC-ES2L, MRI-ESM2-0, NESM3, and UKESM1-0-LL. Area significant against the multi-model spread is hatched in (**a**)–(**c**) identified by Welch's $t$-test ($p < 0.1$).

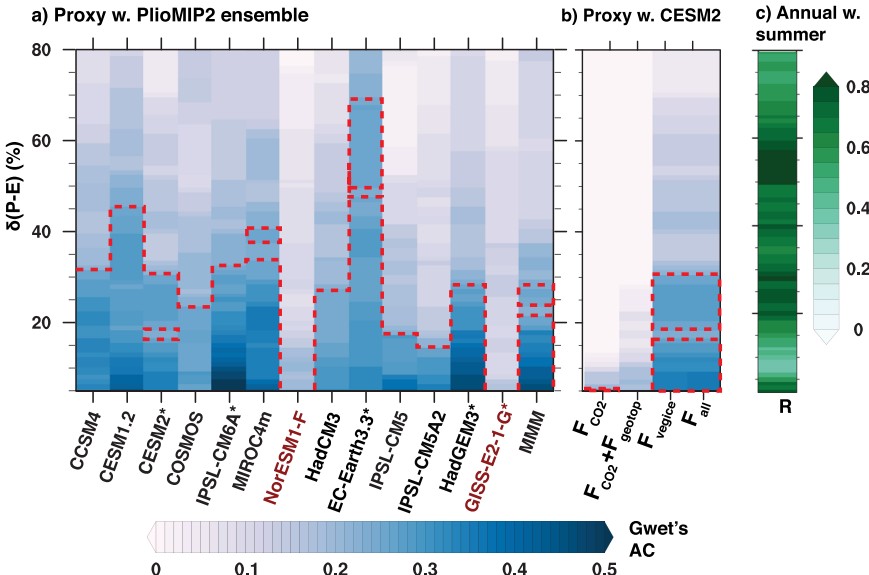

**Fig. 2 Degree of agreement between mid-Pliocene proxy hydroclimate indicators and simulations.** Proxy-model fit is assessed using a measure of categorical agreement between two datasets called Gwet's AC (unitless). For a given % threshold of simulated changes in precipitation minus evaporation ($\delta(P–E)$) relative to the preindustrial, higher (lower) values indicate that proxies and models agree (disagree) that given locations are wet, dry, or neutral. The area within the dashed line indicates statistically significant agreements. **a** Gwet's AC agreement between individual models, Multimodel mean (MMM), and proxies at different thresholds. CMIP6 models are identified with asterisks. **b** Agreement between proxies and simulated $\delta(P–E)$ in response to single or combined climate forcings ($F_{all}$) from changes of $CO_2$ ($F_{CO2}$), tectonics ($F_{geotop}$), and vegetation and ice sheets ($F_{vegice}$). **c** Gwet's AC agreement between annual and boreal summer averages at proxy sites for PlioMIP2 ensemble mean.

IPSL-CM6, and EC-Earth3.3, two models with the highest level of agreement between mid-Pliocene proxies and simulations, show expansive inland wetter conditions across North Africa, Mediterranean, and subtropical East Asia (Fig. S6). In contrast, NorESM-L and GISS-E2-1-G, two of the models with the lowest proxy-model agreement, show highly mixed or muted $\delta(P–E)$ across this region. Similar to the multi-model mean result, the agreement between individual models and proxy hydroclimate indicators is also strongly driven by the simulated summer $\delta(P–E)$ (Fig. S4).

**$CO_2$ or boundary conditions in driving positive $\delta(P–E)$.** Single forcing experiments are commonly used to quantify climate responses to individual forcing agent[42]. In our case, three sets of simulations are constructed with CESM2[39] to quantify contributions to $\delta(P–E)$ from a range of mid-Pliocene forcings: a 400 ppm $CO_2$ ($F_{CO2}$), changes in biome distribution and ice sheets ($F_{vegice}$), and changes in geography and topography ($F_{geotop}$) (Methods). The separation of $F_{vegice}$ and $F_{CO2}$ is designed to separate the influences of vegetation and ice sheet changes from the direct influences of $CO_2$ changes. The former represents Earth system feedbacks, which are not typically considered when evaluating equilibrium climate responses to $CO_2$ forcing[43]. This experimental scheme assumes decorrelation between climate responses to $F_{CO2}$ and $F_{geotop}$, which may not be the case for paleoclimate conditions because various feedbacks have been shown to depend on the background climate warmth at high $CO_2$ levels[44,45]. However, our simulations support this decorrelation under moderate $F_{CO2}$ and $F_{geotop}$. Simulated responses to $F_{geotop}$ are consistent with modern or mid-Pliocene $pCO_2$, and simulated responses to $F_{CO2}$ are consistent with modern or mid-Pliocene topography and geography (Fig. S7). For $F_{vegice}$, land surface and vegetation changes considered here are a consequence of mid-Pliocene $F_{CO2}$ and $F_{geotop}$[43]. Therefore, $F_{vegice}$ is separated from the combined $F_{CO2}$ and $F_{geotop}$.

$F_{vegice}$ explains ~78% (2.2 mm day$^{-1}$) of the regional mean $\delta(P–E)$ induced by $F_{all}$ (2.8 mm day$^{-1}$) across the subtropical

Sahel and East Asia, while contributions from $F_{CO2}$ and $F_{geotop}$ are small (Fig. 1f, g). Furthermore, only $F_{vegice}$ produces a similar level of proxy-model agreement in $\delta(P–E)$ compared to $F_{all}$ (Fig. 2b). Both $F_{CO2}$ and combined $F_{geotop}$ and $F_{CO2}$ produce low values of Gwet's AC (Fig. 2b). The proxy-model agreement seen in the full forcing ($F_{all}$) experiment is only reproducible in the experiment forced with $F_{vegice}$.

Compared to $F_{all}$, $F_{vegice}$ accounts for ~50% of the increase in the global mean surface temperature (2.7 °C) (Fig. 3) and 58% of the increase (0.45 g/kg) in the global mean tropospheric (100 to 1000 hPa) specific humidity. In the Northern subtropics (between 20°N and 30°N), $F_{vegice}$ explains an even greater fraction of tropospheric moistening (61%, or 0.56 g/kg). In contrast, $F_{CO2}$ accounts for 45% of the global mean surface warming (2.4 °C) (Fig. S11), 31% (0.24 g/kg) of the global mean tropospheric moistening, and only 26% of the tropospheric moistening in the northern subtropics. $F_{geotop}$ has much smaller influences on temperature and moisture responses globally. The influence of $F_{geotop}$ is slightly elevated in the northern subtropics accounting for 13% of the increase (0.11 g/kg) in the tropospheric humidity. Warming due to $F_{vegice}$ is attributable to both lowered surface albedo and enhanced evapotranspiration. Areas where boreal forest shifts and expands northward and where mid-latitude deserts become vegetated (Fig. 3a) feature substantially lowered surface albedo (Fig. 3b). Expanded boreal forests also show large increases in upward latent heat flux, suggesting enhanced water vapor feedback[46] (Fig. 3c). A similar increase in latent heat flux also occurs in the northern subtropics where $\delta(P–E)$ is positive.

**The dynamical linkage between climate forcing conditions and mid-Pliocene hydroclimate.** In order to identify the dynamical linkage between climate forcing conditions and hydroclimate in the subtropical Sahel and East Asia, we adapt previously published moisture budget diagnostics (MBD)[47] to decompose simulated June to September $\delta(P–E)$ by individual models (Methods) into changes in the seasonal cycle of tropospheric

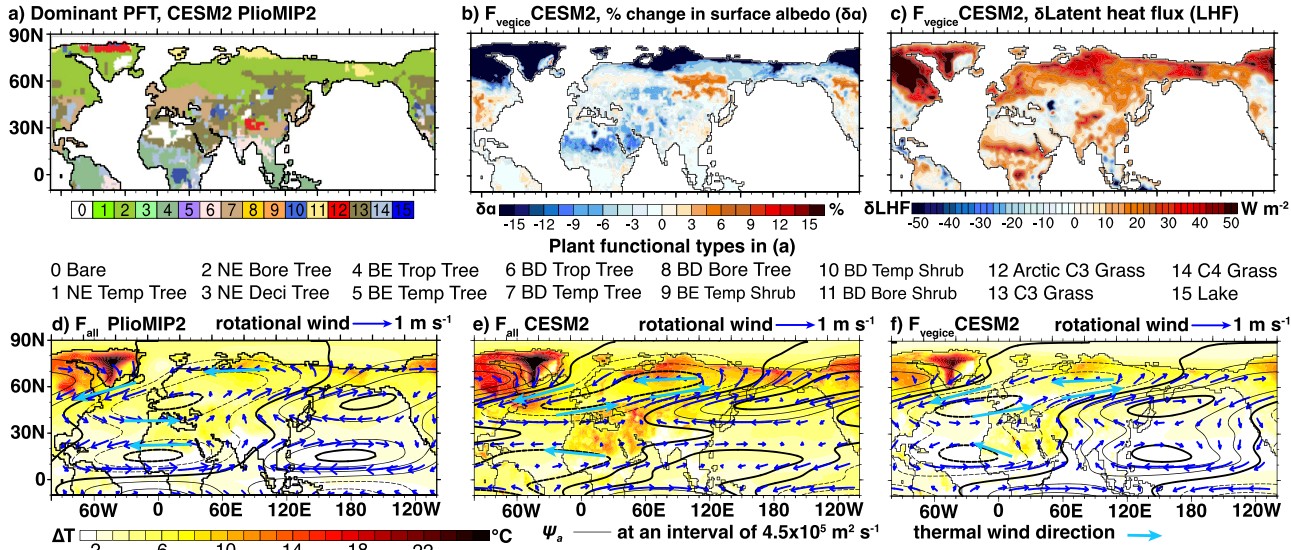

**Fig. 3 Climate responses to the full mid-Pliocene climate forcing ($F_{all}$) and forcing from vegetation and ice sheet changes ($F_{vegice}$).** **a** vegetation and ice sheet distribution: the plant functional type (PFT) with the highest percentage in a grid cell is shown. Changes of **b** surface albedo ($\delta\alpha$, %) and **c** surface upward latent heat flux ($\delta LHF$, W m$^{-2}$) induced by $F_{vegice}$. Label abbreviations in **a** are, NE needleleaf evergreen, BE broadleaf evergreen, BD broadleaf deciduous, Temp temperate, Bore boreal, Deci deciduous, Trop tropical. **d–f** Simulated responses of surface temperature ($\Delta T$, °C) (color shaded), and wave number 1 of the 600 hPa stationary wave ($\psi_a$, contour, m$^2$ s$^{-1}$) and corresponding rotational winds (vectors, m s$^{-1}$) to **d** all forcings ($F_{all}$) in the PlioMIP2 ensemble, **e** $F_{all}$ in CESM2, and **f** $F_{vegice}$ in CESM2. The diagnosed thermal wind directions based on the $\Delta T$ pattern are also shown.

moisture content ($\delta(P–E)_t$), changes in the zonal mean circulation dynamics ($\delta(P–E)_{[V]}$) and stationary wave dynamics ($\delta(P–E)_{V*}$), changes in the tropospheric moisture content ($\delta(P–E)_Q$), changes in the interactions between moisture and circulation dynamics ($\delta(P–E)_{VQ}$), and a residual term ($\delta(P–E)_{RES}$) (Fig. 4 and Fig. S8). The stationary wave response is quantified as the temporal mean departure from the zonal mean following the classic circulation decomposition[48]. The residual term combines the effects of variability of transient eddies and changes of topography (see Methods).

As revealed by MBD, positive $\delta(P–E)$ in the subtropical Sahel and East Asia primarily arises from changes in stationary wave dynamics ($\delta(P–E)_{V*}$) and increased tropospheric moisture content ($\delta(P–E)_Q$) (Fig. 4b, c). Contributions from both $\delta(P–E)_t$ and $\delta(P–E)_{VQ}$ are insignificant (Fig. S8). Intermodel spread in simulated $\delta(P–E)$ strongly scales with the spread in $\delta(P–E)_{V*}$ and $\delta(P–E)_Q$, with little dependency on other terms (Fig. S10). Moreover, the MBD results are consistent between the PlioMIP2 ensemble mean, and CESM2 experiments with $F_{all}$ and $F_{vegice}$ (Fig. 4e, f, and Fig. S8). These results suggest a strong dynamical linkage between $\delta(P–E)_{V*}$, $\delta(P–E)_Q$, and $\delta(P–E)$ across PlioMIP2 models and that this dynamical linkage can mostly be reproduced with $F_{vegice}$.

We further decompose the stationary wave response into components associated with different wave numbers through Fourier decomposition of stream function anomalies ($\psi_a$) at 600 hPa. $\psi_a$ is calculated as departures from the zonal mean and PI (Fig. 3d–f). As shown by the Fourier decomposition, zonal wavenumber 1 of $\psi_a$ accounts for 67% of the total wave energy between 0 and 90°N. This wave component features cyclones separately centered over western Europe and North Africa, and anticyclones centered over the North Pacific (Fig. 3d–f). Following the southern edge of the cyclonic wave centers, a moisture transport corridor emerges: westerly winds bring moisture to North Africa from the tropical Atlantic; southerly and south-westerly winds bring moisture from the tropical Indian and Pacific Oceans towards the Indian subcontinent and East Asia. Furthermore, the rotational winds of this wave component are connected to the surface warming pattern via the thermal wind

relationship. This pattern is broadly reproduced in the simulation forced with $F_{vegice}$ (Fig. 3f).

Positive $\delta(P–E)$ in both regions also results from increased tropospheric moisture content ($\delta(P–E)_Q$). Under PI conditions, low-level winds converge towards North Africa and subtropical East Asia, with the diverging flow in the adjacent regions during the boreal summer (Fig. 4c). This circulation is a key feature of the regional summer monsoon[49]. Even without changing this circulation, elevated tropospheric moisture content can result in greater moisture convergence and positive $\delta(P–E)_Q$ in the subtropical Sahel and East Asia, and greater moisture divergence and negative $\delta(P–E)_Q$ in the adjacent regions. This $\delta(P–E)_Q$ pattern is a known signature of thermodynamic response to elevated $CO_2$, i.e., the wet-gets-wetter, dry-gets-drier paradigm[7,12,50]. As shown in $F_{vegice}$, a similar thermodynamic response can be induced through land cover changes.

## Discussion

In our simulations, mechanisms causing the positive $\delta(P–E)$ in the subtropical Sahel and East Asia are different from previous studies. The MBD reveals that both $\delta(P–E)_{[V]}$ and $\delta(P–E)_{RES}$ are small (Fig. 4a, d). A small $\delta(P–E)_{[V]}$ suggests little contribution from the zonal mean Hadley Circulation change. If Hadley Circulation change were to drive boreal summer $\delta(P–E)$, coherent $\delta(P–E)$ across longitudes would be expected[18]. This is not the case for PlioMIP2 simulations (Fig. 1b, and Fig. S6). Most Plio-MIP2 simulations display a clear latitudinal offset in positive $\delta(P–E)$ between East Asia and West Pacific (Fig. S6) and between Sahel and Central America.

A small $\delta(P–E)_{RES}$ suggests a small net influence from changes of transient eddies and topography on $\delta(P–E)$ (Fig. 4d). Tropical SST warming during the El Niño events influences extratropical pre-cipitation by strengthening and shifting the positions of storm tracks and the subtropical jet equatorward[25]. These changes are not observed in mid-Pliocene simulations of the northern extratropics (Fig. S9). In this area, storm activity weakens as shown by changes in the 850 hPa eddy kinetic energy. The subtropical jet, measured as the

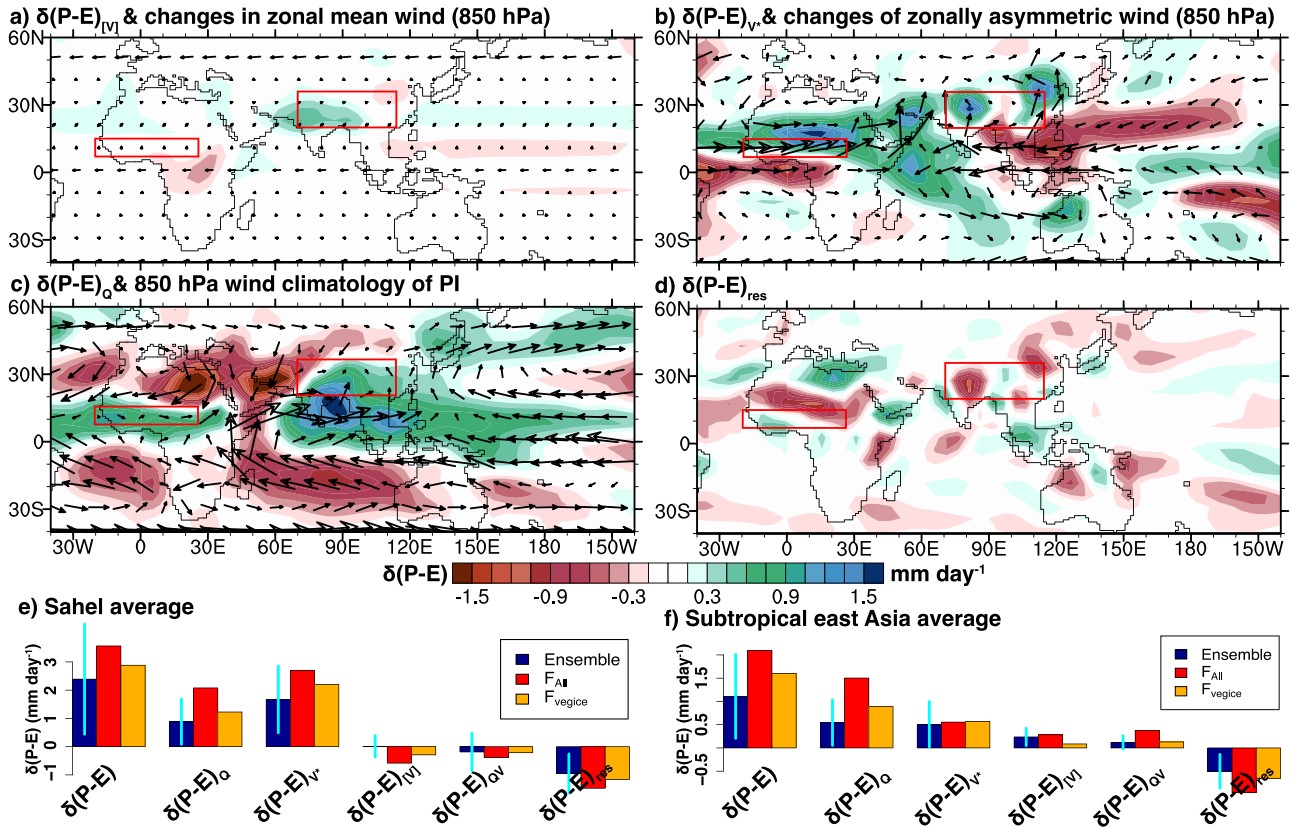

**Fig. 4 Moisture budget diagnostics of mid-Pliocene boreal summer changes in precipitation minus evaporation ($\delta(P–E)$, mm day$^{-1}$). a–d** Contributions to $\delta(P–E)$ from zonal mean circulation ($\delta(P–E)_{[V]}$), changes of stationary wave dynamics ($\delta(P–E)_{V*}$), tropospheric moistening ($\delta(P–E)_Q$), and a residual term($\delta(P–E)_{RES}$). **e, f** Regional mean (red boxes in (**a**)–(**d**)) moisture budget for the subtropical Sahel and East Asia simulated by PlioMIP2 ensemble, and CESM2 with $F_{all}$ and $F_{vegice}$. Error bars show 1 standard deviation of model spread.

200 hPa zonal wind speed, also weakens across Eurasia but remains in a similar position to PI. These changes are consistent with a reduction in the poleward temperature gradient[51] (Fig. S9a, b). Moreover, the pattern of precipitation variability induced by El Niño SSTs[52] has a classic signature of negative $\delta(P–E)$ in North Africa paired with positive $\delta(P–E)$ in Southeast Asia. This signature is not observed in mid-Pliocene simulations: $\delta(P–E)$ is broadly consistent across both regions (Fig. 1b and S6). Models suggest that El Niño-like SSTs, therefore, do not play a significant role in driving mid-Pliocene $\delta(P–E)$ in these regions.

However, $\delta(P–E)_{RES}$ is more noticeable in the North Pacific and subtropical western North America. The impact of El Niño-like SSTs could be important for $\delta(P–E)$ in both regions[13]. Past hydroclimate changes in western North America are shown to be sensitive to changes in eddy moisture transport associated with varying conditions of atmospheric rivers[53–55]. $\delta(P–E)$ changes in this region are likely driven by processes different from the subtropical Sahel and East Asia.

The stationary wave response identified here is distinct from the response of the ITCZ shift. Both the Hadley Circulation change and ITCZ shift highlight the importance of changing the zonal mean energy budget on circulation dynamics[14,18,56–60]. Instead, the stationary wave response in PlioMIP2 experiments is generated by a zonally heterogeneous pattern of energy perturbation and warming. Both are induced by continental greening. Circulation changes in the lower troposphere can be explained by this surface warming pattern through the diagnostic thermal winds, which closely follow the zonal wavenumber 1 of $\psi_a$ (Fig. 3). This "pattern effect" has often been overlooked when examining past hydroclimate changes.

Why is $F_{vegice}$ more effective at altering terrestrial hydroclimate compared to $F_{CO2}$ in the subtropical Sahel and East Asia? Land cover changes are known to be key to generating surface temperature and circulation responses across North Africa and subtropical East Asia[61–63] and may even alter the strength of the Atlantic meridional overturning circulation[64,65]. In North Africa, expansion of desert results in enhanced surface albedo, surface cooling, strengthened diabatic subsidence, and dust emission[66,67]. These responses may further suppress moist convection and perpetuate desert[68–71]. A positive vegetation-precipitation feedback results in multiple equilibrium states of North African vegetation over the late Quaternary[62]. Different from those mechanisms, in our simulations, atmospheric circulation responses to $F_{vegice}$ facilitate moisture transport towards the subtropical Sahel and East Asia via a stationary wave response. This response is closely tied to the large-scale warming pattern generated by $F_{vegice}$, which does not occur in $F_{CO2}$ (Fig. S11). Additionally, this response also differs from the regional wave response driven by diabatic heating from the South Asian monsoon moist convection as suggested by the previous studies[63]. The latter features contrasting wave center and hydroclimate responses between South Asia and the western Sahel, distinct from our findings of a continental-scale wave pattern with consistent responses between these two regions.

Different hydroclimate responses to $F_{CO2}$ and $F_{vegice}$ may also reflect different soil moisture and plant physiological responses. Increasing $CO_2$ may favor the reduction of soil moisture and partitioning of surface heat flux toward sensible heat, leading to enhanced surface warming. This in turn lowers relative humidity above the surface and diminishes continental cloud cover,

contributing to negative $\delta(P-E)$ on land. A similar response of surface heat flux partitioning and moisture deficit can be caused by the reduction of leaf transpiration in response to $CO_2$ fertilization. Soil moisture feedback and $CO_2$ fertilization drive much of the predicted future subtropical terrestrial hydroclimate change[15], and may also contribute to the muted $\delta(P-E)$ response to $F_{CO_2}$ in our experiments despite a modest increase in precipitation. In contrast, $F_{vegice}$ features no direct physiological effect of $CO_2$ and produces large increases in latent heat flux relative to sensible heat flux across continents (Fig. S12), creating favorable conditions for a more humid troposphere. Humidification of the subtropical Sahel and East Asia, therefore, reflect a synergy of dynamic and thermodynamic responses to $F_{vegice}$. Reduced ice sheet cover, expansive northern high-latitude boreal forests, and the vegetated Sahel and Central Asia have all been recorded during other Cenozoic warm intervals[72–74]. These land cover changes were likely instrumental in driving global mean temperature and the terrestrial hydrological cycle throughout the Cenozoic.

Our results suggest an alternate view of the role of vegetation changes in modulating continental hydroclimate. Changes in regional circulation in the form of stationary wave responses lead to strengthened low-level winds that import moisture into the subtropical Sahel and East Asia from the tropical Atlantic and Indian Oceans, respectively. This inland moisture influx is further amplified by enhanced tropospheric humidity. This mechanism is distinct from previous mechanisms that highlight the role of the zonal mean Hadley Circulation or local ecosystem responses to land surface changes. Although complete feedbacks from vegetation and ice sheets are not prognostically simulated in most mid-Pliocene simulations (Supplementary Table 2), the presented simulations highlight radiative perturbations from proxy-constrained changes in vegetation and ice sheets are key to generating hydroclimate changes in the northern subtropics. In comparison, prescribed mid-Pliocene topography and land–sea distribution have little effect (Fig. 1g).

A key implication of our findings is that the mid-Pliocene continental hydroclimate is more appropriately viewed as part of the Earth system feedbacks instead of an immediate response to $F_{CO_2}$, and hence, requires appropriate boundary conditions of vegetation and ice sheet to simulate. This inference is supported by the strong relationship between a model's ability to capture the mid-Pliocene hydroclimate state and its simulated global mean warming (Fig. 5a). The latter reflects model diversity in Earth System Sensitivity (ESS), which incorporates surface temperature responses to long-term biome range shift and ice sheet changes in addition to $CO_2$ changes. In contrast, the relationship between a model's ability to capture the mid-Pliocene hydroclimate state and its equilibrium climate sensitivity (ECS) is weak (Fig. 5b). ECS of an individual model is estimated from the doubling $CO_2$ experiment with the same model and resolution and without biome range and ice sheet changes[33]. Published studies have mostly focused on the impact of Earth system feedback on temperature responses[43,75]. Here, we demonstrate that these feedbacks are key for understanding paleohydrological responses. Inconsistencies in interpretations of proxy hydroclimate records[18,76] may be resolved by considering $F_{vegice}$.

Lastly, our results offer a resolution to the apparent discrepancies between the projections of hydroclimate changes in the Sahel and subtropical East Asia following the middle-of-the-road scenarios and strong geologic evidence for moist Pliocene climate at similar levels of $CO_2$. While the near-future is dominated by the short-term response to $CO_2$ radiative forcing and internal variability, Pliocene hydroclimate reflects long-term adjustments of the Earth system that incorporates responses from vegetation and ice sheets, which occur on timescales longer than most future climate projections. Feedbacks of vegetation and ice sheets to $CO_2$ increase are known to amplify the response of

equilibrium surface temperature to radiative forcing[43,75]. We highlight that these relatively slow Earth system feedbacks are also critical for understanding Earth's hydroclimate responses to varying $CO_2$. Therefore, changes in vegetation and ice sheet distributions should be carefully considered when simulating past and future climates.

## Methods

**Proxy-model comparison**. Our proxy compilation builds on previous efforts to compile Pliocene hydroclimate records. These include the compilations created by refs. [18,40,41]. The records included in these sources span a variety of proxy types, from sedimentological indicators of lacustrine environments, palynological indicators of vegetation composition, faunal remains, and stable isotope records from organic and inorganic materials. We added to these compilations by including new records of terrestrial hydroclimate dating to the Pliocene archived on the NOAA Paleoclimatology Database (https://www.ncdc.noaa.gov/data-access/paleoclimatology-data) and Pangaea Database (https://www.pangaea.de/).

To identify the average hydroclimate signal during the mid-Pliocene, we filtered available records based on the precision of their age models. Specifically, proxy records were required to include at least two age control points, with one age control point after the mid-Pliocene and a clearly identifiable age control point prior to mid-Piacenzian. For many records, this "basal" tie point was often in the early Pliocene or Miocene. Only including records with multiple age control points reduces the likelihood that samples inferred to be from the mid-Pliocene actually come from earlier in the Pliocene or the early Pleistocene. For most proxy records, age control derives from a combination of magnetostratigraphy (e.g., the Gauss–Matuyama boundary), radiometric dating, or biostratigraphic information. In some cases, identification of independently-dated tephra layers or correlation with the benthic oxygen isotope stack provides age control. These filtering criteria allowed us to retain a compilation of 62 records. Of the 62 records included, 30 come from paleobiological indicators like faunal remains or pollen, while the other 34 records are drawn from interpretations of sedimentary sequences or stable isotopic analyses of organic and inorganic materials (SI). A supplementary excel file (Supplementary Data 1) with details on the proxy records, including methods, chronology, and original citation, is included with this manuscript. Because of the broad, continental-scale coherence of the model signal, proxy-model synthesis results are similar when using the compilations found in other sources e.g., ref. [41].

We rely on the author's original interpretation about whether the record reflects, on average, wetter or drier conditions, or no change in hydroclimate during the mid-Pliocene compared to late Quaternary/modern conditions. Many records are discontinuous, and cannot provide quantitative comparisons between mid-Pliocene climate and late quaternary or PI conditions, and low-resolution records do not resolve climate cycles within the mid-Pliocene. Our proxy data, therefore, represent comparisons of the change between average mid-Pliocene conditions and late Holocene or modern conditions. In contrast, PlioMIP2 modeling experiments are designed to target a particular orbital interval during the Pliocene (MIS KM5c) with present-day orbital forcings[37]. Hence, the impact of orbital forcing is not included by PlioMIP2 experiments by design. The influence of orbital forcing on hydroclimate conditions is not recorded by most proxy records due to limited temporal resolution and dating accuracy of these records. Moreover, several high-resolution, continuous hydroclimate records from North Africa have shown relatively small hydroclimate variability during the late Pliocene[77–79]. Keeping in mind this limitation, our analyses focus on qualitatively assessing the agreement between proxies and models. Despite this limitation, our compilation shows similar large-scale hydroclimate features compared to previous Pliocene hydroclimate synthesis efforts, notably, wetter conditions evidenced by widespread lakes, reduced dust flux, and more mesic environments across southern and eastern Asia, North Africa, and parts of Europe. For proxy-model comparison, we include records spanning 0–67°N and 23°W–172°E. Details of broad regional trends are discussed in the SI.

**Statistical analysis**. Because the majority of these records have qualitative or semi-quantitative interpretations, we assess the fit between proxies and models qualitatively. We classify modeled mid-Pliocene $\delta(P-E)$ as indicating wetter, drier, or no change relative to PI simulations from the same model, and we classify proxies as indicating wetter, drier, or no change based on the author's original interpretation of mid-Pliocene vs. present-day or late Quaternary conditions. To convert continuous, quantitative model outputs of $\delta(P-E)$ into categorical data, we classify model output as showing wetter, drier, or no change at different thresholds of % change in $P-E$. For instance, choosing a 10% threshold, we would classify models that show more than a 10% increase in $P-E$ as showing wetter conditions for a particular site. We vary the threshold at 1% intervals between 1 and 100% change and separately calculate proxy-model agreement.

The degree of proxy-model agreement was assessed using Gwet's AC statistic, a metric of inter-rater agreement similar to Cohen's kappa ($\kappa$). While the latter has been used in previous proxy-model comparison studies, the nature of the Pliocene proxy data, where the vast majority of records show wetter conditions, renders Cohen's $\kappa$ statistic less robust[80]. Cohen's $\kappa$ is known to perform poorly as a result of Cohen's paradox whereby the statistic underestimates the true agreement between

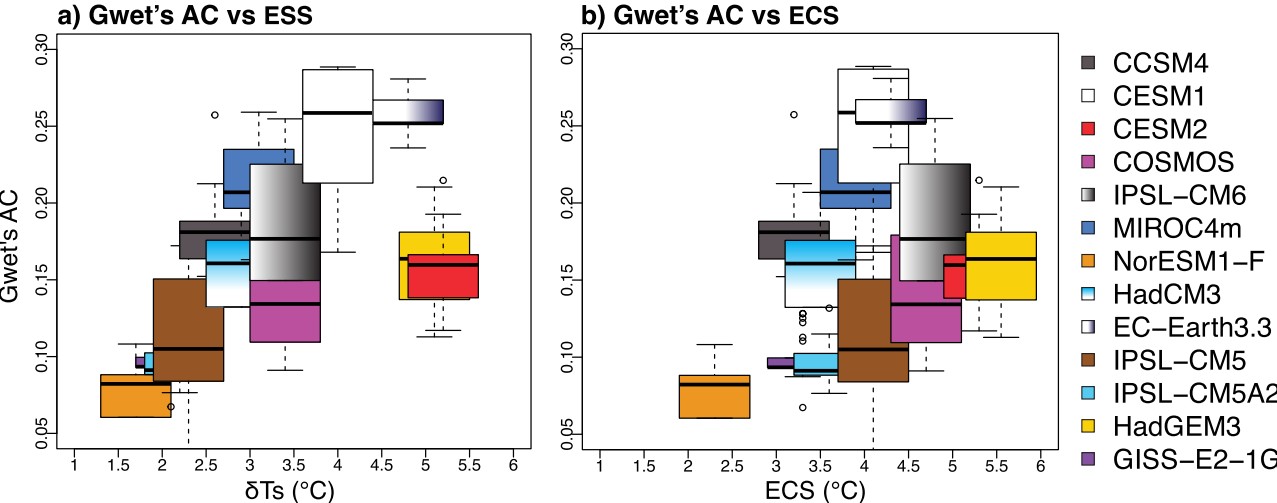

**Fig. 5 Relationship between earth system sensitivity (ESS, °C), equilibrium climate sensitivity (ECS, °C), and ability to reproduce mid-Pliocene proxy hydroclimate state by different models.** Proxy data-model agreement measured by Gwet's ACs as a function of **a** global mean surface warming ($\delta T_s$, which quantifies ESSs), and **b** ECSs of these models. $\delta T_s$ and ECSs are from ref. [33]. Boxplots show the spread of Gwet's AC (1st, 2nd, and 3rd quantiles, upper and lower fences, and outliers if any) from using 20–60% of $\delta(P-E)$ as thresholds (Method).

raters in cases of skewed distributions of ratings across categories[80]. We, therefore, use a related metric known as Gwet's AC, which is known to be resistant to this paradox. Gwet's AC statistic is similar to Cohen's in that the AC statistic is

$$AC = \frac{P_a - P_{exp}}{100\% - P_{exp}}$$

Where $P_a$ is the actual percentage of agreement between proxies and models (e.g., the percentage of wetter sites that are correctly classified by the model as wetter, the percentage of drier sites correctly classified by the model, etc.), and $P_{exp}$ reflects the expected percentage of agreement between raters (e.g., proxies and models) by chance alone. The Gwet's AC statistic is identical to Cohen's $\kappa$ but uses a different formulation for $P_{exp}$ that is not susceptible to Cohen's paradox. The statistical significance of Gwet's AC statistic was calculated according to the error estimator and methods outlined in ref. [80]. Results of statistical significance testing are presented in Fig. S3.

To avoid weighting our analysis towards regions with a greater density of proxy records less than 150 km that featured the same sign of the change (e.g., both showing wetter, drier, or no change) from each other were combined into one site. However, in cases with records showing opposite signs of change, we retain both records since in many cases there is not enough information to determine which record is more reliable, and excluding both would decrease the data coverage of our proxy compilation in key regions like East Asia (Fig. S2).

In the absence of a priori evidence that proxies reflect hydroclimate in a particular season, patterns of the proxy-model agreement are assessed using annually averaged $P-E$. We independently assess the fit between proxies and models for winter (December–February) and summer (June–September) separately (Fig. S4). The fit is much higher across all models and the multi-model mean for June–September rainfall. Proxy-model agreement for winter rainfall alone shows very low values of Gwet's AC statistic, suggesting that winter rainfall changes do not explain a significant component of the pattern seen in the proxy record. Furthermore, intermodel spread in Gwet's AC values on annually averaged rainfall show a strong correlation with intermodel Gwet's AC results calculated for June-September $P-E$ (Fig. 2) at nearly all threshold values of $P-E$. This suggests that the summer seasonal signal, which is also the largest signal across model simulations (Fig. 1), drives the overall pattern of proxy-model agreement across models.

**PlioMIP2 ensemble**. We use a suite of model simulations conducted as part of the 2nd Pliocene Model Intercomparison Project (PlioMIP2)[35,81–87]. Boundary conditions are derived from the PRISM4 dataset[38]. PRISM4 boundary conditions include information on land distributions, topography and bathymetry, vegetation, soils, lakes, and land ice cover. Two experimental protocols were developed, one implementing Pliocene conditions with a modern land/sea mask, and the enhanced experiment that included all boundary conditions[37,38]. $pCO_2$ is prescribed at 400 ppm. Other trace gases, orbital parameters, and solar output were set to be identical to the PI control simulation of each model.

Modeling groups were given the option to either prescribe vegetation changes or simulate vegetation changes using a dynamic global vegetation model. For the latter experiment, model simulations were started with PI vegetation and the model was allowed to spin up until a new equilibrium distribution of vegetation was achieved. We provide basic information on the configuration of each of the PlioMIP2 models used in our analyses in Supplementary Table 2. This information is also provided in the previous study[33]. We note that only one modeling group in the suite of

simulations we analyzed opted to use a dynamic configuration for vegetation, suggesting that the choice of dynamic vs. static vegetation is not the primary source of spread across model results. We also note that models show varying levels of agreement with the signal in proxies (Fig. 2), suggesting that our results are not trivial (e.g., models with prescribed vegetation reproduce said vegetation) since the spatial pattern of hydroclimate in each model appears to depend on model design.

**Sensitivity experiments using community earth system model version 2 (CESM2)**. Despite the overall similarity in geography and topography to present-day, mid-Pliocene boundary conditions feature several changes in ocean gateways, islands, and lake distributions that may influence regional climate[38,41,88–90]. PlioMIP2 simulations also feature expanded grassland replacing the subtropical desert of North Africa and Central Asia and afforestation at northern high latitudes as well as deglaciated western Antarctic and most of the Greenland[38].

To isolate the mechanisms responsible for Pliocene hydroclimate changes, we carried out several new experiments with CESM2 that decompose simulated responses to the full mid-Pliocene climate forcing ($F_{all}$) into responses to forcings from elevated $CO_2$ ($F_{CO2}$), changes in paleo-geography and -topography ($F_{geotop}$), and changes in biome distribution and ice sheets ($F_{vegice}$). We also constrain potential state dependency of responses induced by $F_{CO2}$ and $F_{geotop}$. Because the forcing from $F_{vegice}$ only emerges as a result of long-term $CO_2$ changes and a fixed $F_{geotop}$ condition, we build this state dependency into the experimental approach. These new experiments separately feature (1) a 400 ppm $CO_2$ and PI geography and topography (E400); (2) PI vegetation and ice sheets, but otherwise mid-Pliocene $CO_2$, geography and topography (Eo400), (3) mid-Pliocene geography and topography, but PI $CO_2$, and vegetation and ice sheets (Eo280). The design and naming convention of these simulations roughly follow the Tier II PlioMIP2 protocol[37]. All simulations were run with $0.9 \times 1.6°$ resolution for atmosphere and land, and 1° nominal resolution for ocean and sea ice components, resulting in ~100 km resolution of all model components.

These new simulations are carried out for a minimum of 300 model years. Diagnostics of equilibrium by the global mean top of the atmosphere radiation imbalance and surface temperature are shown in Fig. S13. To produce climatology, we average the last 100 years of the model simulation. Eo400, E400, Eo280 together with the published full forcing (Eoi400) and PI experiments (E280) allow the decomposition of climate responses (denoted as R) to $F_{all}$ into the sum of responses to individual forcings: $R(F_{all}) = R(F_{vegice}) + R(F_{geotop}) + R(F_{CO2})$, for which $R(F_{all}) = R(Eoi400) - R(E280)$; $R(F_{geotop}) = R(Eo400) - R(E400)$ or $R(F_{geotop}) = R(Eo280) - R(E280)$; $R(F_{CO2}) = R(E400) - R(E280)$ or $R(F_{CO2}) = R(Eo400) - R(Eo280)$; $R(F_{vegice}) = R(Eoi400) - R(Eo400)$. The comparison of $R(F_{geotop})$, estimated with mid-Pliocene and PI levels of $pCO_2$, quantifies the dependency of $R(F_{geotop})$ on the background $pCO_2$. Similarly, the comparison of $R(F_{CO2})$, estimated with mid-Pliocene and PI geography and topography, quantifies the dependency of $R(F_{CO2})$ on the background geography and topography (Fig. S7).

**Development of moisture budget diagnostics**. To diagnose the causes of changes in terrestrial water balance, measured by changes in precipitation (P) minus evaporation (E) and referred to as $\delta(P-E)$, we further developed and applied the moisture budget diagnostics[47] to PlioMIP2 simulations. With only monthly data available, our derivation aims to facilitate the comparison of a pair of experiments and the evaluation of existing hypotheses regarding the cause of mid-Pliocene hydroclimate

change (e.g., identifying whether mid-Pliocene hydroclimate anomalies results from zonal mean changes or stationary wave changes).

Following Eq. (13) in ref. [47] on pressure coordinates, $P$–$E$ is balanced by changes in the moisture tendency and moisture convergence

$$P - E = -\frac{1}{g\rho_w}\frac{\partial}{\partial t}\int_0^{P_s} q\,dp - \frac{1}{g\rho_w}\nabla\cdot\int_0^{P_s} \boldsymbol{u}q\,dp \qquad (1)$$

In Eq. (1), $g$ is geopotential acceleration, $\rho_w$ is the density of water, $P_s$ is surface pressure, $q$ is specific humidity, $p$ is pressure, and $\boldsymbol{u}$ is horizontal wind. For a pair of experiments, experiment 1 is the control case (e.g., PI) and experiment 2 is the sensitivity experiment (e.g., Pliocene), the small perturbation method tells us that $q_2 = q_1 + \delta q$; $\boldsymbol{u_2} = \boldsymbol{u_1} + \delta\boldsymbol{u}$; $(P-E)_2 = (P-E)_1 + \delta(P-E)$; and $P_{s2} = P_{s1} + \delta P_s$. The experiment (1 or 2) is identified by the subscript. $\delta$ terms are the small perturbations. We can decompose the perturbation to $P$–$E$ into contributions from the anomalous moisture tendency, changes in convergence due to changes in wind, moisture, and the interaction of these changes, as well as a residual term, which are shown on the right side of Eq. (2) from the left to the right:

$$\delta(P-E) = -\frac{1}{g\rho_w}\frac{\partial}{\partial t}\int_0^{P_{s1}} \delta q\,dp - \frac{1}{g\rho_w}\nabla\cdot\int_0^{P_{s1}} (q_1\delta\boldsymbol{u} + \boldsymbol{u_1}\delta q + \delta\boldsymbol{u}\delta q)\,dp + \text{res1} \qquad (2)$$

$$\text{res1} = -\frac{1}{g\rho_w}\frac{\partial}{\partial t}q_2\delta P_s - \frac{1}{g\rho_w}\nabla\cdot(\boldsymbol{u_2}q_2\delta P_s) \qquad (3)$$

Residual term 1 (RES 1, Eq. 3) quantifies influences from changing surface pressure, which is dominated by topographic changes between the mid-Pliocene and PI. Apply Reynold's decomposition, we can separate the total change of a given quantity into its temporal mean (e.g., monthly or annual climatology), denoted via an overbar ($\bar{\phantom{x}}$), and higher frequency temporal fluctuations, denoted via a prime ($'$). Such that

$$\delta q = \delta\bar{q} + \delta q', \delta\boldsymbol{u} = \delta\bar{\boldsymbol{u}} + \delta\boldsymbol{u}' \qquad (4)$$

$$\delta(P-E) = \overline{\delta(P-E)} + \delta(P-E)', \boldsymbol{u_1} = \bar{\boldsymbol{u_1}} + \boldsymbol{u_1}', q_1 = \bar{q_1} + q_1', P_{s1} = \bar{P}_{s1} + P_s' \qquad (5)$$

Given that we are interested in understanding the contributions from zonal mean circulation changes compared to other changes, we further separate $\delta\bar{\boldsymbol{u}}$ into changes in zonal mean, indicated by square brackets ([]) and changes in deviations from the zonal mean (i.e., the stationary wave), which is indicated by an asterisk (*)

$$\delta\bar{\boldsymbol{u}} = [\delta\bar{\boldsymbol{u}}] + \delta\bar{\boldsymbol{u}}^*$$

Using these expansions, we can rephrase the anomalous moisture budget in Eq. (2) to separate out the contributions due to zonal mean and stationary waves. The updated form of Eq. 2 is

$$\overline{\delta(P-E)} = -\frac{1}{g\rho_w}\frac{\partial}{\partial t}\int_0^{\bar{P}_{s1}} \overline{\delta q}\,dp - \frac{1}{g\rho_w}\nabla\cdot\int_0^{\bar{P}_{s1}} (\bar{\boldsymbol{u_1}}\overline{\delta q} + \bar{q_1}([\delta\bar{\boldsymbol{u}}] + \delta\bar{\boldsymbol{u}}^*) + \delta\bar{\boldsymbol{u}}\overline{\delta q})\,dp + \text{res1} + \text{res2} \qquad (6)$$

Residual term 2 (Eq. 7) quantifies combined effects of transient eddies moisture transport and influxes from the surface:

$$\text{res2} = -\frac{1}{g\rho_w}\frac{\partial}{\partial t}\overline{\delta q' P_s'} + \frac{1}{g\rho_w}\nabla\cdot[\overline{(\bar{\boldsymbol{u_1}}\delta q' + \boldsymbol{u_1}'\overline{\delta q} + q_1'\delta\bar{\boldsymbol{u}} + \bar{q_1}\delta\boldsymbol{u}' + \delta\boldsymbol{u}'\overline{\delta q} + \delta\bar{\boldsymbol{u}}\delta q')P_s'}]$$
$$-\frac{1}{g\rho_w}\nabla\cdot\int_0^{\bar{P}_{s1}} \overline{(\delta q'\delta\boldsymbol{u}' + q_1'\delta\boldsymbol{u}' + \boldsymbol{u_1}'\delta q')}\,dp \qquad (7)$$

Calculating residual terms 1 and 2 require high-frequency outputs of surface pressure, three-dimensional specific humidity, and horizontal winds, which are not available. Also, even at the 6-hourly resolution, the calculated eddy terms are insufficient to close the moisture budget with reanalysis data[47]. Thereby, RES 1 and RES 2 are not explicitly calculated in our calculations and quantified as a combined residual (i.e., the difference between $P$–$E$ changes and the sum of other terms in the moisture budget equation). Yet, the decomposition demonstrates physical interpretations of these residuals. Monthly climatologies of surface pressure, precipitation, evaporation, three-dimensional horizontal winds, and specific humidity are used to calculate the remaining terms in Eq. 6. From the left to right, the first term in Eq. 6 ($-\frac{1}{g\rho_w}\frac{\partial}{\partial t}\int_0^{\bar{P}_{s1}}\overline{\delta q}\,dp$) is the moisture tendency term, which quantifies contributions to $\overline{\delta(P-E)}$ from changes in seasonal cycle. The term $-\frac{1}{g\rho_w}\nabla\cdot\int_0^{\bar{P}_{s1}}(\bar{\boldsymbol{u_1}}\overline{\delta q})\,dp$ describes contributions from changes in climatological mean tropospheric moisture content; $-\frac{1}{g\rho_w}\nabla\cdot\int_0^{\bar{P}_{s1}}(\bar{q_1}[\delta\bar{\boldsymbol{u}}])\,dp$ describes contributions from changes in the zonal mean circulation; $-\frac{1}{g\rho_w}\nabla\cdot\int_0^{\bar{P}_{s1}}(\bar{q_1}\delta\bar{\boldsymbol{u}}^*)\,dp$ describes contributions from changes in stationary wave kinetics. Finally, $-\frac{1}{g\rho_w}\nabla\cdot\int_0^{\bar{P}_{s1}}(\delta\bar{\boldsymbol{u}}\overline{\delta q})\,dp$ describes contributions from covarying changes in mean moisture content and horizontal circulation.

## Data availability

The PlioMIP2 simulations analyzed in this study that are part of the Climate Model Intercomparison Project 6 (CMIP6) have been deposited to the Earth System Grid Federation with the access link: https://doi.org/10.22033/ESGF/CMIP6.4804 for EC-Earth3, https://doi.org/10.22033/ESGF/CMIP6.5230 for IPSL-CM6A-LR, https://doi.org/10.22033/ESGF/CMIP6.7227 for GISS-E2-1-G, https://doi.org/10.22033/ESGF/CMIP6.7675 for CESM2, and https://doi.org/10.22033/ESGF/CMIP6.12130 for HadGEM. The processed multi-model ensemble of PlioMIP2 and CESM2 data are available at Zenedo with the access link: https://zenodo.org/record/5706370#.YZbc6y1h0eY.

## Code availability

Source code of CESM2[39] can be downloaded from https://escomp.github.io/CESM/versions/cesm2.1/html/downloading_cesm.html.

The code for moisture budget analysis[91] is published at Zenedo with the access link: https://zenodo.org/record/5706370#.YZbc6y1h0eY

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

## Acknowledgements

The authors would like to thank all modeling groups who provided PMIP4 outputs for this analysis, WCRP, CMIP panel, PCMDI, ESGF infrastructures for sharing data, WCRP, and CLIVAR for supporting the PMIP project. R.F., T.B., B.L.O., and E.C.B acknowledge support from U.S. National Science Foundation grant numbers 1814029 and 1903650 (R.F.), 1903148 and 2103015 (T.B.) and 1852977 (B.L.O. and E.C.B). X.L. and N.T. acknowledge support from the National Science Foundation of China grant numbers 42005042 (X.L.) and 41888101 (N.T.). D.L. acknowledges support from NERC (Natural Environment Research Council), SWEET Large Grant number NE/P01903X/1. C.C. acknowledges support from France ANR HADoC grant number ANR-17-CE31-0010. Q.Z. acknowledge support from Swedish Research Council (Vetenskapsrådet) grant numbers. 2013-06476 and 2017-04232. W.L.C. and A.A.O. acknowledge support from Japanese JSPS Kakenhi grant 17H06104 and NEXT Kakenhi grant 17H06323. H.D. acknowledges support from USGS Paleoclimate Research and Development Program. C.S. and G.L. acknowledge funding via the Alfred Wegener Institute's research programme PACES2. C.S. received funding via the Helmholtz Climate Initiative REKLIM. The PRISM4 reconstruction and boundary conditions used in the presented simulations were funded by the U.S. Geological Survey Climate and Land Use Change Research and Development Program. Any use of trade, firm, or product names is for descriptive purposes only and does not imply endorsement by the U.S. Government. The CESM2 simulations are performed with high-performance computing support from Cheyenne (doi:10.5065/D6RX99HX) provided by NCAR's Computational and Information Systems Laboratory, sponsored by the National Science Foundation. The IPSL-CM6A-LR simulation was run on the Très Grande Infrastructure de Calcul (TGCC) at Commissariat à l'Energie Atomique (gencmip6 project) under the allocations 2016-A0030107732, 2017-R0040110492 and 2018-R0040110492 (project gencmip6) provided by GENCI (Grand Equipement National de Calcul Intensif). The model simulations with EC-Earth3 and the data analysis were performed using resources provided by ECMWF's computing and the Swedish National Infrastructure for Computing (SNIC) at the National Supercomputer Centre (NSC), which is partially funded by the Swedish Research Council through grant agreement no. 2018-05973. A.A.O. and W.L.C. acknowledge JAMSTEC for use of the Earth Simulator supercomputer. COSMOS simulations have been conducted at the Computing and Data Centre of the Alfred Wegener Institute – Helmholtz Centre for Polar and Marine Research on a NEC SX-ACE high-performance vector computer.

## Author contributions

R.F., T.B. B.L.O.-B., and A.M.H. conceptualized this study. R.F., T.B., M.K., and D.L. devised the methods for analysis. R.F. and T.B. led the investigation of the question, compiled data, and implemented the methods. R.F. and T.B. wrote the original draft. H.D. contributed data to create boundary conditions of PlioMIP2 simulations. E.C.B., A.H., J.T., S.H., A.A.-O., W.-L.C., M.K., C.C., C.G., X.L., G.L., C.S., N.T., Q.Z., Z.Z., Z.H., C.J.R.W., D.J.L., D.C., and W.R.P. contributed simulation data to this study, and reviewed and edited various versions of this paper.

## Competing interests

The authors declare no competing interests.
