## [Peer Review File · Nature Communications]

REVIEWER COMMENTS

Reviewer #1 (Remarks to the Author):

In the manuscript “Past terrestrial hydroclimate sensitivity controlled by Earth System Feedbacks” by Feng et al., the authors analyze output from a suite of model simulations conducted as part of PlioMIP2, performed individual forcing experiments using CESM2, and compiled proxy data indicative of past hydrological changes, all in an attempt to assess the driving factors behind patterns in the mid-Pliocene terrestrial water balance, with a focus on two specific regions: the Sahel and subtropical East Asia. By combining these approaches, the authors show that the hydroclimate state of the mid-Pliocene is predominantly influenced by long-term feedbacks in the Earth System related to CO₂ forcing of ice volume and vegetation, and that the direct radiative effects of atmospheric CO₂ are secondary. The effects of geographic and topographic changes on mid-Pliocene hydrological conditions are minor. The differences in land cover related to ice sheets and vegetation drive elevated P-E in the Sahel and subtropical East Asia during the mid-Pliocene, specifically related to the stationary wave response of spatial variability in surface warming as well as enhanced tropospheric humidity. Overall, these findings emphasize the importance of long-term feedbacks to atmospheric CO₂ forcing related to terrestrial hydroclimate.

The findings in this manuscript have broad impacts for both the modern and past climate communities, specifically those who focus on the hydrological cycle as well as short- and long-term Earth System feedbacks related to atmospheric CO₂. I find the explanation of the methodology to be quite thorough and straightforward, and I especially appreciate the critical selection of the available proxy data. All this being said, I think there are some improvements that could be made to the manuscript, specifically dealing with the ‘Results’ and ‘Discussion’. In addition, I think the manuscript needs to be checked for grammatical issues and consistency. Overall, I believe this work could represent an important finding that would be suited for publication in Nature Communications if these concerns are adequately addressed.

Major Comments:

1) The frequency of grammatical errors and inconsistency of term usage detracts from the flow of the manuscript. While the issues are not a big deal in any one instance, I did find myself stopping and re-reading sentences often because of a missing “the” or “a”, or a change in the way “East Asia” was written (just as some examples). Because I am not well-versed in the modeling techniques or moisture budget diagnostics utilized in this work, and thus am unable to substantially comment on that component of the manuscript, I am providing several suggestions for grammatical fixes (in addition to the few other comments I have below). I do believe this is an important addition to the existing work on Pliocene climate, and a solid manuscript overall, which is why I think the writing needs to be clearer.

2) I think the discussion on the observed differences between the results from the PlioMIP2 model ensemble mean and the expected responses to El Niño-like SSTs needs to be broken out into its own discussion section, not quickly brought up as it is currently in L211-221. This is one of the two current hypotheses invoked to explain enhanced subtropical moisture during the mid-Pliocene, and if it is to be argued against, needs to be done in detail. I am in no way saying I disagree with the observations provided, I would just like to see the lines of argumentation expanded and separated out. It may be beneficial to put the discussion of results related to the El Niño-like SSTs and the Hadley Circulation hypotheses into their own section, which could reside either in the 'Results' or 'Discussion'.

3) The few sentences on desertification feedbacks related to dust (lines 263-265) require citations, specifically for the sentence suggesting "These responses may further suppress moist convection and perpetuate desert.". Some potential citations specifically related to dust and its effect on precipitation that would suffice include:

Rosenfeld, D., Rudich, Y., & Lahav, R. (2001). Desert dust suppressing precipitation: A possible desertification feedback loop. *Proceedings of the National Academy of Sciences*, 98(11), 5975-5980.

Yoshioka, M., Mahowald, N. M., Conley, A. J., Collins, W. D., Fillmore, D. W., Zender, C. S., & Coleman, D. B. (2007). Impact of desert dust radiative forcing on Sahel precipitation: Relative importance of dust compared to sea surface temperature variations, vegetation changes, and greenhouse gas warming. *Journal of Climate*, 20(8), 1445-1467.

Min, Q. L., Li, R., Lin, B., Joseph, E., Wang, S., Hu, Y., ... & Chang, F. (2009). Evidence of mineral dust altering cloud microphysics and precipitation. *Atmospheric Chemistry and Physics*, 9(9), 3223-3231.

Zhao, C., Liu, X., Ruby Leung, L., & Hagos, S. (2011). Radiative impact of mineral dust on monsoon precipitation variability over West Africa. *Atmospheric Chemistry and Physics*, 11(5), 1879-1893.

4) Due to my lack of expertise, I cannot provide a critical evaluation of the modeling and $\delta(P-E)$ decomposition approaches used in this work. However, I find the descriptions of the methods to be quite detailed and straightforward. I also appreciate the rigorous approach the authors took in selecting proxy data, and for making the moisture budget code available.

Minor Comments:

L39-44: This sentence has unnecessary words in some parts, and seems to be missing some in others. Also, it is quite long, which makes it difficult to follow. One suggestion would be:

“Despite muted tectonic changes and atmospheric CO₂ levels (pCO₂) similar to present-day, geological reconstructions from the mid-Pliocene (MP, 3.3-3.0 Ma) indicate drastic reorganizations of subtropical terrestrial hydroclimate during this interval. Specifically, there is evidence for high lake levels in Northern Africa and for mesic conditions in subtropical Eurasia, suggesting a moist subtropical terrestrial hydroclimate across the Sahel and subtropical Eurasia compared to today.”

L44-45: Maybe change to “based on a proxy data compilation” or “a compilation of proxy data”.

L57-59: The references “Schuster et al., 2009” and “Wang et al., 2019” are not references 1 and 2, respectively, nor are they found anywhere in the reference list. References 1 and 2 in the ‘References’ section do fit the information provided in lines 57-59, so it seems as if the “Schuster et al., 2009” and “Wang et al., 2019) references need to be added to the reference list and the numbers adjusted accordingly.

L60: I am unsure why the reference for this part of the sentence is to the ‘Method’ section. I would suggest adding an actual reference here or stating in those parentheses why the reader should be looking at the ‘Methods’.

L63: “Could be either “A moderate increase of precipitation” or “Moderate increases of precipitation”.

L69-70: I think this should be “with strong modulations from land warming patterns, land-sea warming contrast, tropical SST patterns”.

L72: To be consistent with the usage earlier in the ‘Introduction’, I would suggest “across the subtropical Sahel and East Asia”. This is a good example of where the usage of terms (in this case “the subtropical Sahel and East Asia”) get switched around throughout the manuscript. I would suggest checking all instances where these locations, as well as “North Africa”, are discussed and make sure of three things:

1) There is a “the” in front of “subtropical Sahel” or “Sahel”, unless it is being used to describe a specific component of the Sahel, such as “Sahel hydroclimate”.

2) “East Asia” is capitalized.

3) "North Africa" is capitalized.

L72: "relative to both the historical period"

L75: "SSP2-4.5".

L76: I think this should be "between the hydroclimate state"

L94-95: Change to "SST patterns"

L100: I think this should be "We focus on these regions".

L101: May be better to use "the coherence" instead of "its coherence".

L105: I would suggest changing part of this sentence to "wetter MP conditions in the Sahel and East Asia are driven".

L134: "(see Methods).".

L164-165: These two sentences need to be rewritten and possibly combined. I am unsure what the authors are saying here.

L166: Should this be "For Fvegice"?

L175: Should this be changed to "accounts for ~50% of the increase in global mean surface temperature" and "and 58% of the increase in global mean tropospheric"?

L177: "between 20°N and 30°N"

L179: I would suggest saying "and only 26%".

L182: "accounting for 13% of the increase".

L187: Should be "A similar increase in latent".

L194: "into changes in the seasonal cycle".

L199: The citation here is in the wrong format.

L201: I would change to "The MBD results are consistent between the PlioMIP2 ensemble mean and Fvegice.".

L209-210: Should be "the North American continent."

L215: Change to "and the subtropical jet stream".

L219-220: Use either "in northern Africa" or "in North Africa" and "in southeast Asia".

L231: I would change this to "cyclones separately centered over western Europe and North Africa, and anticyclones centered over the North Pacific"

L235: "the tropical Indian and Pacific Oceans towards the Indian subcontinent and East Asia.".

L240: Is this the correct figure reference? Maybe Fig. 4c?

L284-285: This sentence is confusing, mostly due to the phrasing: "moist subtropical Sahel and East Asia". Are the authors saying, "a moister subtropical Sahel and East Asia"?

L288-289: "and the terrestrial hydrological cycle throughout the Cenozoic.".

L295: Should be "the tropical Atlantic and Indian Oceans".

L312: "of the hydrological cycle".

L320: “by the short-term response”.

Figure Comments:

Figure 1:

L607: Should be “Shared Socioeconomic Pathway 2-4.5”.

L608-609: I suggest changing this sentence to “(b) – (c) Pliocene annual and boreal summer (June to September) mean $\delta(P-E)$ of PlioMIP2 ensemble. Proxy data displayed as circles.”.

L610: Should be “between the subtropical Sahel (10° – 20°N, 10°W – 25°E) and East Asia”.

L611: Should be “SSP2-4.5”.

L615: Should be “SSP2-4.5”.

L618-619: Should say “the difference between the Pliocene and preindustrial simulations.”.

Figure 2:

L630-631: I do not think the description for part (c) is correct. It seems to only be indicative of the multi-model mean(?).

Figure 4:

L643-646: I do not think the order of the listed components matches up with the order of the subplots (a-d).

Supplement Comments:

L82: The section should be separated out as the others were above.

L195: This reference should be “deMenocal, P. B.,”.

Reviewer #2 (Remarks to the Author):

Review of NCOMMS-21-35067 by Chris Brierley

This is a very impressive piece of work. The findings are quite important and represent a significant advance in our understanding about Pliocene climates specifically, and future hydroclimate in general. I strongly recommend publication. There is an awful lot of research that has gone into this work. It provides a new proxy compilation, multiple new climate model simulations, and a complicated analysis framework.

One unfortunate consequence is that fitting so much science into a single short manuscript makes for a rather dense read. I don't think this can (or should) be avoided, and would not be dismayed to see the text published as is. If there were going to be revisions, the authors might want to consider these suggestions that might make it easier to follow:

- Could the abbreviation “MP” just be replaced by a single word – maybe “Pliocene”?
- L63-67. Is this describing the future or past?
- Do you really need to abbreviate Hadley circulation to HC?
- L115. This sentence adds little and doesn't act as a good introduction to the results section
- I appreciate that $\delta(P-E)$ is the correct way of writing precipitation minus evaporation but it reads awkwardly – especially when combined with other text brackets. Is there an alternative
- L115-152 has more references to supplementary figures than actual ones.
- I got lost as to what “wave-1 of SF” means on L257

I was intrigued that precession wasn't mentioned anywhere. Is that a confounding factor in the proxy reconstructions? I wouldn't expect more than a sentence on this question in the manuscript, as I suspect it can never be fully discounted.

- Fig 1a should use the same spatial resolution as the rest of the panels in Fig 1.

- I feel that an "all" column is needed in Fig 2b, even though it would only replicate the CESM2 column.

Response to reviewers

We thank both reviewers for their constructive comments and suggestions. The original comments are in black. Our responses are in blue. The revised manuscript with tracked changes is appended at the end.

Reviewer #1 (Remarks to the Author):

In the manuscript “Past terrestrial hydroclimate sensitivity controlled by Earth System Feedbacks” by Feng et al., the authors analyze output from a suite of model simulations conducted as part of PlioMIP2, performed individual forcing experiments using CESM2, and compiled proxy data indicative of past hydrological changes, all in an attempt to assess the driving factors behind patterns in the mid-Pliocene terrestrial water balance, with a focus on two specific regions: the Sahel and subtropical East Asia. By combining these approaches, the authors show that the hydroclimate state of the mid-Pliocene is predominantly influenced by long-term feedbacks in the Earth System related to CO₂ forcing of ice volume and vegetation, and that the direct radiative effects of atmospheric CO₂ are secondary. The effects of geographic and topographic changes on mid-Pliocene hydrological conditions are minor. The differences in land cover related to ice sheets and vegetation drive elevated P-E in the Sahel and subtropical East Asia during the mid-Pliocene, specifically related to the stationary wave response of spatial variability in surface warming as well as enhanced tropospheric humidity. Overall, these findings emphasize the importance of long-term feedbacks to atmospheric CO₂ forcing related to terrestrial hydroclimate.

The findings in this manuscript have broad impacts for both the modern and past climate communities, specifically those who focus on the hydrological cycle as well as short- and long-term Earth System feedbacks related to atmospheric CO₂. I find the explanation of the methodology to be quite thorough and straightforward, and I especially appreciate the critical selection of the available proxy data. All this being said, I think there are some improvements that could be made to the manuscript, specifically dealing with the ‘Results’ and ‘Discussion’. In addition, I think the manuscript needs to be checked for grammatical issues and consistency. Overall, I believe this work could represent an important finding that would be suited for publication in Nature Communications if these concerns are adequately addressed.

We thank the reviewer for their interest and their thoughtful comments. We have revised the manuscript according to these suggestions. We have also corrected the grammatical errors and inconsistencies in wording throughout the manuscript.

Major Comments:

1) The frequency of grammatical errors and inconsistency of term usage detracts from the flow of the manuscript. While the issues are not a big deal in any one instance, I did find myself stopping and re-reading sentences often because of a missing “the” or “a”, or a change in the way “East Asia” was written (just as some examples). Because I am not well-versed in the

modeling techniques or moisture budget diagnostics utilized in this work, and thus am unable to substantially comment on that component of the manuscript, I am providing several suggestions for grammatical fixes (in addition to the few other comments I have below). I do believe this is an important addition to the existing work on Pliocene climate, and a solid manuscript overall, which is why I think the writing needs to be clearer.

Thank you for the careful read through of the MS. We have now thoroughly checked the manuscript and corrected several grammatical errors and inconsistencies in wording.

2) I think the discussion on the observed differences between the results from the PlioMIP2 model ensemble mean and the expected responses to El Niño-like SSTs needs to be broken out into its own discussion section, not quickly brought up as it is currently in L211-221. This is one of the two current hypotheses invoked to explain enhanced subtropical moisture during the mid-Pliocene, and if it is to be argued against, needs to be done in detail. I am in no way saying I disagree with the observations provided, I would just like to see the lines of argumentation expanded and separated out. It may be beneficial to put the discussion of results related to the El Niño-like SSTs and the Hadley Circulation hypotheses into their own section, which could reside either in the 'Results' or 'Discussion'.

In the revised manuscript, the discussion of influences of Hadley Circulation and El Niño-like SSTs on simulated mid-Pliocene hydroclimate is organized into its own section (Line 231 to Line 257). For the discussion of the influences of Hadley Circulation, we also highlighted results from Fig. S6, which demonstrates that simulated $\delta(P-E)$ is asymmetric across longitudes. This asymmetry cannot be explained by a change in the zonal-mean Hadley Circulation. For the discussion about the influences of El Niño-like SSTs, we highlight that changes in the jet stream, storm track activities, and the spatial pattern of $\delta(P-E)$ do not match signatures of influences of El Niño-like SSTs (line 239 and 251).

3) The few sentences on desertification feedbacks related to dust (lines 263-265) require citations, specifically for the sentence suggesting "These responses may further suppress moist convection and perpetuate desert.". Some potential citations specifically related to dust and its effect on precipitation that would suffice include:

Rosenfeld, D., Rudich, Y., & Lahav, R. (2001). Desert dust suppressing precipitation: A possible desertification feedback loop. *Proceedings of the National Academy of Sciences*, 98(11), 5975-5980.

Yoshioka, M., Mahowald, N. M., Conley, A. J., Collins, W. D., Fillmore, D. W., Zender, C. S., & Coleman, D. B. (2007). Impact of desert dust radiative forcing on Sahel precipitation: Relative importance of dust compared to sea surface temperature variations, vegetation changes, and greenhouse gas warming. *Journal of Climate*, 20(8), 1445-1467.

Min, Q. L., Li, R., Lin, B., Joseph, E., Wang, S., Hu, Y., ... & Chang, F. (2009). Evidence of mineral dust altering cloud microphysics and precipitation. *Atmospheric Chemistry and Physics*, 9(9),

3223-3231.

Zhao, C., Liu, X., Ruby Leung, L., & Hagos, S. (2011). Radiative impact of mineral dust on monsoon precipitation variability over West Africa. *Atmospheric Chemistry and Physics*, 11(5), 1879-1893.

Thanks very much for providing these references. We incorporated them in the revised manuscript (line 273).

4) Due to my lack of expertise, I cannot provide a critical evaluation of the modeling and $\delta(P-E)$ decomposition approaches used in this work. However, I find the descriptions of the methods to be quite detailed and straightforward. I also appreciate the rigorous approach the authors took in selecting proxy data, and for making the moisture budget code available.

Thank you for these comments.

Minor Comments:

L39-44: This sentence has unnecessary words in some parts, and seems to be missing some in others. Also, it is quite long, which makes it difficult to follow. One suggestion would be:

“Despite muted tectonic changes and atmospheric CO₂ levels (pCO₂) similar to present-day, geological reconstructions from the mid-Pliocene (MP, 3.3-3.0 Ma) indicate drastic reorganizations of subtropical terrestrial hydroclimate during this interval. Specifically, there is evidence for high lake levels in Northern Africa and for mesic conditions in subtropical Eurasia, suggesting a moist subtropical terrestrial hydroclimate across the Sahel and subtropical Eurasia compared to today.”

We thank the reviewer for this suggestion. This section of the abstract is rewritten as “Despite tectonic conditions and atmospheric CO₂ levels (pCO₂) like present-day, geological reconstructions from the mid-Pliocene (3.3-3.0 Ma) document high lake levels in the Sahel and mesic conditions in subtropical Eurasia, suggesting drastic reorganizations of subtropical terrestrial hydroclimate during this interval.”

L44-45: Maybe change to “based on a proxy data compilation” or “a compilation of proxy data”.

This is now corrected to the latter phrasing.

L57-59: The references “Schuster et al., 2009” and “Wang et al., 2019” are not references 1 and 2, respectively, nor are they found anywhere in the reference list. References 1 and 2 in the ‘References’ section do fit the information provided in lines 57-59, so it seems as if the “Schuster et al., 2009” and “Wang et al., 2019) references need to be added to the reference list and the numbers adjusted accordingly.

These reference inconsistencies have now been corrected. We added the following two references in the list and changed the reference numbers accordingly.

Wang, H., Lu, H., Zhao, L., Zhang, H., Lei, F. and Wang, Y., 2019. Asian monsoon rainfall variation during the Pliocene forced by global temperature change. *Nature communications*, 10(1), pp.1-8.

Schuster, M., Düringer, P., Ghienne, J.F., Roquin, C., Sepulchre, P., Moussa, A., Lebatard, A.E., Mackaye, H.T., Likius, A., Vignaud, P. and Brunet, M., 2009. Chad Basin: paleoenvironments of the Sahara since the Late Miocene. *Comptes Rendus Geoscience*, 341(8-9), pp.603-611.

L60: I am unsure why the reference for this part of the sentence is to the 'Method' section. I would suggest adding an actual reference here or stating in those parentheses why the reader should be looking at the 'Methods'.

This statement was drawn from a compilation of proxy data described in supplementary Table S1. This table was referenced in the revised manuscript.

L63: "Could be either "A moderate increase of precipitation" or "Moderate increases of precipitation".

This has been changed to "Moderate increases of precipitation".

L69-70: I think this should be "with strong modulations from land warming patterns, land-sea warming contrast, tropical SST patterns".

This has now been corrected.

L72: To be consistent with the usage earlier in the 'Introduction', I would suggest "across the subtropical Sahel and East Asia". This is a good example of where the usage of terms (in this case "the subtropical Sahel and East Asia") get switched around throughout the manuscript. I would suggest checking all instances where these locations, as well as "North Africa", are discussed and make sure of three things:

- 1) There is a "the" in front of "subtropical Sahel" or "Sahel", unless it is being used to describe a specific component of the Sahel, such as "Sahel hydroclimate".
- 2) "East Asia" is capitalized.
- 3) "North Africa" is capitalized.

We thank the reviewer for the careful read! We searched the manuscript and made all corrections according to your suggestions.

L72: "relative to both the historical period"

This has now been corrected.

L75: “SSP2-4.5”.

This has now been corrected.

L76: I think this should be “between the hydroclimate state”

This has now been corrected.

L94-95: Change to “SST patterns”

This has now been corrected.

L100: I think this should be “We focus on these regions”.

This has now been corrected.

L101: May be better to use “the coherence” instead of “its coherence”.

This has now been corrected.

L105: I would suggest changing part of this sentence to “wetter MP conditions in the Sahel and East Asia are driven”.

This has now been corrected.

L134: “(see Methods).”.

This has now been corrected.

L164-165: These two sentences need to be rewritten and possibly combined. I am unsure what the authors are saying here.

This has now been corrected. These two sentences are rewritten as “simulated responses to F_{geotop} are consistent with modern or mid-Pliocene $p\text{CO}_2$, and simulated responses to F_{CO_2} are consistent with modern or mid-Pliocene topography and geography” (Line 161 to Line 163).

L166: Should this be “For F_{vegice} ”?

This has now been corrected.

L175: Should this be changed to “accounts for ~50% of the increase in global mean surface temperature” and “and 58% of the increase in global mean tropospheric”?

This has now been corrected.

L177: “between 20°N and 30°N”

This has now been corrected.

L179: I would suggest saying “and only 26%”.

This has now been corrected.

L182: “accounting for 13% of the increase”.

This has now been corrected.

L187: Should be “A similar increase in latent”.

This has now been corrected.

L194: “into changes in the seasonal cycle”.

This has now been corrected.

L199: The citation here is in the wrong format.

This has now been corrected.

L201: I would change to “The MBD results are consistent between the PlioMIP2 ensemble mean and Fvegice.”

This has now been corrected.

L209-210: Should be “the North American continent.”

This has now been corrected.

L215: Change to “and the subtropical jet stream”.

This has now been corrected.

L219-220: Use either “in northern Africa” or “in North Africa” and “in southeast Asia”.

This has now been corrected.

L231: I would change this to “cyclones separately centered over western Europe and North Africa, and anticyclones centered over the North Pacific”

This has now been corrected.

L235: “the tropical Indian and Pacific Oceans towards the Indian subcontinent and East Asia.”

This has now been corrected.

L240: Is this the correct figure reference? Maybe Fig. 4c?

This has now been corrected. It is Fig. 4c.

L284-285: This sentence is confusing, mostly due to the phrasing: “moist subtropical Sahel and East Asia”. Are the authors saying, “a moister subtropical Sahel and East Asia”?

This sentence has been rewritten as “Humidification of the subtropical Sahel and East Asia reflects a synergy of dynamic and thermodynamic responses to Fvegice.”

L288-289: “and the terrestrial hydrological cycle throughout the Cenozoic.”

This has now been corrected.

L295: Should be “the tropical Atlantic and Indian Oceans”

This has now been corrected.

L312: “of the hydrological cycle”.

This has now been corrected.

L320: “by the short-term response”.

This has now been corrected.

Figure Comments:

Figure 1:

L607: Should be “Shared Socioeconomic Pathway 2-4.5”.

This has now been corrected.

L608-609: I suggest changing this sentence to “(b) – (c) Pliocene annual and boreal summer (June to September) mean $\delta(P-E)$ of PlioMIP2 ensemble. Proxy data displayed as circles.”.

This has been changed according to this suggestion.

L610: Should be “between the subtropical Sahel ($10^{\circ} - 20^{\circ}N$, $10^{\circ}W - 25^{\circ}E$) and East Asia”.

This has been changed according to this suggestion.

L611: Should be “SSP2-4.5”.

This has now been corrected.

L615: Should be “SSP2-4.5”.

This has now been corrected.

L618-619: Should say “the difference between the Pliocene and preindustrial simulations.”.

This has now been corrected.

Figure 2:

L630-631: I do not think the description for part (c) is correct. It seems to only be indicative of the multi-model mean(?).

Yes, it does refer to the MMM. This has now been corrected.

Figure 4:

L643-646: I do not think the order of the listed components matches up with the order of the subplots (a-d).

This has now been corrected.

Supplement Comments:

L82: The section should be separated out as the others were above.
This has now been corrected.

L195: This reference should be “deMenocal, P. B.,”.
This has now been corrected.

Reviewer #2 (Remarks to the Author):

Review of NCOMMS-21-35067 by Chris Brierley

This is a very impressive piece of work. The findings are quite important and represent a significant advance in our understanding about Pliocene climates specifically, and future hydroclimate in general. I strongly recommend publication. There is an awful lot of research that has gone into this work. It provides a new proxy compilation, multiple new climate model simulations, and a complicated analysis framework.

One unfortunate consequence is that fitting so much science into a single short manuscript makes for a rather dense read. I don't think this can (or should) be avoided, and would not be dismayed to see the text published as is. If there were going to be revisions, the authors might want to consider these suggestions that might make it easier to follow:

- Could the abbreviation “MP” just be replaced by a single word – maybe “Pliocene”?

We thank the reviewer for the support for our study. This is a great point. We replaced all the “MP” with mid-Pliocene in the revised manuscript. Given that the chosen pCO_2 and boundary conditions for PlioMIP2, and the age window filter for compiling proxy hydroclimate records both target the mid-Pliocene, we decided to keep the framing as the “mid-Pliocene”.

- L63-67. Is this describing the future or past?

These lines describe the general responses of increasing precipitation in subtropical continents due to increasing CO_2 (Line 61 to 63). This has now been clarified as “Elevated pCO_2 can lead to moderate increases in precipitation across these regions as a result of tropospheric moistening, enhanced land-sea thermal contrast, and enhanced inland moisture advection.”

- Do you really need to abbreviate Hadley circulation to HC?

We have now spelled out all HCs throughout the manuscript.

- L115. This sentence adds little and doesn't act as a good introduction to the results section

We have now removed this sentence in the revised manuscript (originally last sentence in the introduction).

- I appreciate that $\delta(P-E)$ is the correct way of writing precipitation minus evaporation but it reads awkwardly – especially when combined with other text brackets. Is there an alternative.

We rewrote this sentence. It now reads “Mid-Pliocene changes in precipitation minus evaporation, referred to as $\delta(P-E)$, and other climate variables are calculated with respect to preindustrial (PI) values”.

- L115-152 has more references to supplementary figures than actual ones.

We have attempted to limit references to supplemental figures in this section.

- I got lost as to what “wave-1 of SF” means on L257

We thank the reviewer for catching this. We wrote out those abbreviations in the revised manuscript. Stream function is represented by the Greek letter ψ . The zonal wave number 1 is written out in all occasions.

I was intrigued that precession wasn't mentioned anywhere. Is that a confounding factor in the proxy reconstructions? I wouldn't expect more than a sentence on this question in the manuscript, as I suspect it can never be fully discounted.

This compilation of proxy hydroclimate indicators do not contain orbital variability. We added a sentence in the introduction to clarify our interpretations of this data compilation as qualitative or semi-quantitative indicators of mean hydroclimate state given the lack of orbital-scale variabilities documented by these records (Line 127 to 129). More discussion about this point is in the Method section (lines 378 to 381).

- Fig 1a should use the same spatial resolution as the rest of the panels in Fig 1.

This has now been corrected.

- I feel that an “all” column is needed in Fig 2b, even though it would only replicate the CESM2 column.

This panel has now been added to the figure as suggested.

Past terrestrial hydroclimate sensitivity controlled by Earth System Feedbacks

Authors

R. Feng^{1*}, T. Bhattacharya², B. Otto-Bliesner³, E. Brady³, A. Haywood⁴, J. Tindall⁴, S. Hunter⁴, A. Abe-Ouchi⁵, W.-L. Chan⁵, M. Kageyama⁶, C. Contoux⁶, C. Guo⁷, X. Li⁸, G. Lohmann⁹, C. Stepanek⁹, N. Tan¹⁰, Q. Zhang¹¹, Z. Zhang⁸, Z. Han¹², C. J. R. Williams¹³, D. J. Lunt¹³, H. Dowsett¹⁴, D. Chandan¹⁵, W. R. Peltier¹⁵

Affiliations

1. Department of Geosciences, College of Liberal Arts and Sciences, University of Connecticut, Connecticut, USA

2. Department of Earth and Environmental Sciences, Syracuse University

3. Climate and Global Dynamics Laboratory, National Center for Atmospheric Research, Boulder, Colorado, USA

4. School of Earth and Environment, University of Leeds, Woodhouse Lane, Leeds, West Yorkshire, LS29JT, UK

5. Atmosphere and Ocean Research Institute, University of Tokyo, Kashiwa, Japan.

6. Laboratoire des Sciences du Climat et de l'Environnement, LSCE/IPSL, CEA-CNRS-UVSQ, Université Paris-Saclay, F-91191 Gif-sur-Yvette, France

7. NORCE Norwegian Research Centre, Bjerknes Centre for Climate Research, 5007 Bergen, Norway

8. Department of Atmospheric Science, School of Environmental studies, China University of Geoscience, Wuhan, 430074, China

9. Alfred Wegener Institute-Helmholtz Centre for Polar and Marine Research, Bremerhaven, Germany

10. Key Laboratory of Cenozoic Geology and Environment, Institute of Geology and Geophysics, Chinese Academy of Sciences, Beijing 100029, China

11. Department of Physical Geography and Bolin Centre for Climate Research, Stockholm University, Stockholm, Sweden

12. College of Oceanography/Key Laboratory of Marine Hazards Forecasting, Ministry of Natural Resources/Key Laboratory of Ministry of Education for Coastal Disaster and Protection, Hohai University, Nanjing, China

13. School of Geographical Sciences and Cabot Institute, University of Bristol, University Road, Bristol, BS8 1SS, UK

14. Florence Bascom Geoscience Center, U. S. Geological Survey, Reston, Virginia, USA

15. Department of Physics, University of Toronto, Toronto, CA

*Corresponding author. Email: ran.feng@uconn.edu

Abstract

Despite tectonic conditions and atmospheric CO₂ levels (pCO₂) like present-day, geological reconstructions from the mid-Pliocene (3.3-3.0 Ma) document high lake levels in the Sahel and mesic conditions in subtropical Eurasia, suggesting drastic reorganizations of subtropical terrestrial hydroclimate during this interval. Here, using a compilation of proxy data and multi-model paleoclimate simulations, we show that the mid-Pliocene hydroclimate state is not driven by direct CO₂ radiative forcing but by a loss of northern high-latitude ice sheets and continental greening. These ice sheet and vegetation changes are long-term Earth system feedbacks to elevated pCO₂. Further, the moist conditions in the Sahel and

Commented [FR1]: edited in response to reviewer 1's suggestions

Commented [FR2]: edited in response to reviewer 1's suggestions

subtropical Eurasia during the mid-Pliocene are a product of enhanced tropospheric
humidity and a stationary wave response to the surface warming pattern, which varies
strongly with land cover changes. These findings highlight the potential for amplified
terrestrial hydroclimate response over long timescales to a sustained CO₂ forcing.

Introduction

Geologic evidence suggests dramatic reorganizations of subtropical climate during past
greenhouse climate intervals, including the mid-Piacenzian Warm Period (3.3 to 3.0 Ma,
commonly referred to as the mid-Pliocene). Multiple proxies of hydroclimate indicate large, deep
lakes and reduced dust flux across North Africa during the late Pliocene^{1,2}, and more mesic
vegetation in South and East Asia^{3,4}. Additional sedimentary and paleobotanical data also points
to wetter subtropical Eurasia conditions prior to the intensification of northern hemisphere
glaciation (Supplementary Table 1). These changes imply large increases in precipitation minus
evaporation (P-E). All are associated with pCO₂ levels of approximately 400 ppm^{5,6}, similar to
today's level. Elevated pCO₂ can lead to moderate increases in precipitation across these regions
as a result of tropospheric moistening⁷, enhanced land-sea thermal contrast⁸, and enhanced inland
moisture advection^{9,10}. However, evaporation also increases due to surface warming. As a result,
predicted changes in terrestrial water balance (P-E), and associated changes in soil moisture and
runoff remain equivocal across subtropical continents¹¹.

Modeled hydroclimate responses to CO₂ forcing broadly follows the “wet-gets-wetter, dry-
gets-drier” paradigm⁷ with strong modulations from land warming patterns, the land-sea warming
contrast¹², tropical SST patterns¹³, and feedbacks from soil moisture and CO₂ fertilization effects
on leaf phenology^{14,15}. In simulations featuring middle-of-the-road future warming scenarios,
predicted changes in P-E by the end of 21st century (2081 – 2100) are minimal across the
subtropical Sahel and East Asia relative to both the historical period (1986 – 2005) (Fig. 1a and d)
and preindustrial (Supplementary Figure 1). These future scenarios are often thought to be
comparable to the mid-Pliocene climate^{16,17} (Fig. 1a). Sources for the disparity between the
hydroclimate state recorded by mid-Pliocene proxies and future simulations are unknown. One
potential source might be the transient nature of future climate change as opposed to the quasi-
equilibrium nature of mid-Pliocene climate. Yet, even equilibrium simulations with only the mid-
Pliocene CO₂ forcing fail to produce moist subtropical terrestrial conditions¹⁸. Our simulation also
confirms this result (Fig. 1).

From the perspective of atmospheric dynamics, two leading hypotheses have been
proposed to explain the wetter subtropical continents during past warm climates. Both hypotheses
were proposed when many older-generation models at lower spatial resolutions were not able to
simulate wetter subtropical continents. One hypothesis emphasizes the hydroclimate impact of an
El Niño-like Pacific mean state. SST records from the tropical Pacific record greater warming
across the eastern equatorial Pacific than the western Pacific warm pool¹⁹⁻²³, resulting in an El-
Niño-like pattern of SST anomalies. An El Niño-like Pacific SST pattern may strengthen and shift
the subtropical jet equatorward, enhancing the transient eddy-driven moisture convergence and
ascent^{24,25}. The other hypothesis focuses on the role of a sluggish Hadley Circulation that results
from a relaxed meridional SST gradient²⁶⁻²⁸. A weaker Hadley Circulation may result in weakened
zonal mean moisture divergence from the subtropics and, in turn, reduced aridity^{18,29,30}.

With advancements in model development and boundary conditions, newer earth system
model simulations show substantial differences in simulated past climate states^{23,31,32}, particularly
in mid-Pliocene SST patterns³³, polar warmth³⁴, Atlantic Overturning Circulation³⁵, and

Commented [FR3]: The last sentence in the introduction is removed according to reviewer 2

Commented [FR4]:

Commented [FR5]: All occurrences of East Asia are capitalized

Commented [FR6]: Reference corrected (reviewer 1's comment)

Commented [FR7]: Corrected referencing to Supplementary Table 1 (reviewer 1's comment)

Commented [FR8]: rephrased according to the comment of reviewer 2

Commented [FR9]: corrected according to reviewer 1

Commented [FR10]: rewritten according to reviewer 1's suggestion

Commented [FR11]: added “the” throughout the manuscript when appropriate according to Reviewer 1

Commented [FR12]: corrected according to reviewer 1

Commented [FR13]: Spelled out according to reviewer 2

precipitation³⁶. Here, using atmosphere-ocean coupled global climate model (GCM) simulations
from the newest Pliocene Model Intercomparison Project Phase II (PlioMIP2)^{33,37,38}, we
demonstrate that most of the current generation ESMs can reproduce the pattern of mid-Pliocene
hydroclimate of the subtropical Sahel and East Asia suggested by proxies without any
paleoclimate-specific changes to model parameterizations. We focus on these regions given the
large, coherent signal across the majority of PlioMIP2 simulations and the convergence between
proxy data and model simulations (Fig. 1b and 1c). We further develop several new simulations
using the Community Earth System Model version 2^{35,39} to explore the extent to which simulated
mid-Pliocene hydroclimate changes across the subtropical Sahel and East Asia can be generated
by changes in CO₂ radiative forcing, tectonics, or vegetation and ice sheets. In contrast to previous
hypotheses, **wetter mid-Pliocene conditions in the subtropical Sahel and East Asia are driven by**
**tropospheric moistening and changes to stationary wave dynamics in response to surface warming**
**patterns that result from vegetation and ice sheet changes. Both changes are part of the long-term**
**Earth system feedbacks to a sustained CO₂ forcing. Moreover, model skill at simulating Pliocene**
**hydroclimate states strongly scales with Earth System Sensitivities (ESS) of individual models**
**instead of the Equilibrium Climate Sensitivities (ECS).**

Commented [FR14]: corrected according to reviewer 1

Results

P-E pattern in models and proxies

The last 100 years of simulations by 13 PlioMIP2 GCMs (Supplementary Table 2) were
averaged to produce the ensemble mean. **Mid-Pliocene changes in precipitation minus evaporation,**
**referred to as $\delta(P-E)$, and other climate variables are calculated with respect to preindustrial (PI)**
**values. A robust moistening signal that is larger than the intermodel variability is found across the**
**Sahel and subtropical Eurasia (Fig. 1b). This pattern is most pronounced during the boreal summer**
**months (June to September) (Fig. 1c), with little or opposite changes during the boreal winter**
**months (December to March) (Fig. S2). The spatial continuity of positive $\delta(P-E)$ from North Africa**
**to subtropical East Asia is not a visual coincidence: models that show a large precipitation increase**
**in the Sahel also tend to show a large precipitation increase in the subtropical East Asia (Fig. 1d),**
**suggesting similar processes driving hydroclimate changes in both regions. We confirm this using**
**moisture budget analysis (see below).**

Commented [FR15]: rephrased according to the comment of reviewer 2

To compare modeled patterns of hydroclimate change to available geologic data, we
compiled proxy indicators of mid-Pliocene terrestrial hydroclimate, drawing on existing
compilations^{18,40,41} as well as our own literature search. We identify a total of 64 proxy records
that include sedimentological indicators, palynological, floral or faunal, offshore marine records,
and stable isotope analyses of organic and inorganic materials (Table S1 and Fig. S3). **These**
**records are interpreted as qualitative or semi-quantitative indicators of mean hydroclimate state**
**given the lack of orbital-scale variabilities documented in these records (see Methods). We**
**quantify the extent to which proxies and models produce the same patterns of wetter, drier, or**
**unchanged mid-Pliocene hydroclimate changes using a metric known as Gwet's AC designed for**
**categorical data (see Methods). To account for the unknown sensitivity of proxy hydroclimate**
**indicators to Pliocene (P-E) changes, model values of P-E are expressed as % changes from pre-**
**industrial values at the proxy sites.**

Commented [FR16]: stating out the lack of orbital variabilities documented by the records as suggested by Reviewer 2

The annual pattern of $\delta(P-E)$ revealed by the PlioMIP2 ensemble mean shows statistically
significant agreement with proxy indicators of hydroclimate across subtropical Sahel and East Asia
for a wide range of $\delta(P-E)$ thresholds (Fig. 2a). This agreement is strongly driven by the summer
$\delta(P-E)$ pattern (Fig. 2c). Further, a subset of experiments featuring dynamic phenology and

terrestrial carbon cycle also show strong coupling between the positive $\delta(P-E)$ and enhanced net
primarily productivity across both regions, consistent with paleoecological reconstructions (Fig.
S5). Yet, different models show varying skills at capturing the pattern in the proxy records (Figs.
2a). For instance, IPSL-CM6, and EC-Earth3.3, two models with the highest level of agreement
between mid-Pliocene proxies and simulations, show expansive inland wetter conditions across
the North Africa, Mediterranean, and subtropical East Asia (Figure S6). In contrast, NorESM-L
and GISS-E2-1-G, two of the models with the lowest proxy-model agreement, show highly mixed
or muted $\delta(P-E)$ across this region. Similar to the multi-model mean result, the agreement between
individual models and proxy hydroclimate indicators is also strongly driven by the simulated
summer $\delta(P-E)$ (Fig. S4).

**CO₂ or boundary conditions in driving positive $\delta(P-E)$**

Single forcing experiments are commonly used to identify responses to individual climate
forcings⁴². In our case, three sets of simulations are constructed with CESM2³⁹ to quantify
contributions to $\delta(P-E)$ from a range of mid-Pliocene forcings: a 400 ppm CO₂ (F_{CO_2}), changes in
biome distribution and ice sheets (F_{vegice}), and changes in geography and topography (F_{geotop})
(Methods). Separation of F_{vegice} and F_{CO_2} is designed to separate influences of vegetation and ice
sheet changes from the direct influences of CO₂ changes. The former represents Earth system
feedbacks, which are not typically considered when evaluating equilibrium climate responses to
CO₂ forcing⁴³. This experimental scheme assumes decorrelation between the climate responses to
F_{CO_2} and F_{geotop} , which may not be the case for paleoclimate conditions because feedbacks have
been shown to depend on the background climate warmth^{44,45}. However, our simulations support
this decorrelation under moderate F_{CO_2} and F_{geotop} , given that simulated responses to F_{geotop} are
consistent with modern or mid-Pliocene pCO_2 , and simulated responses to F_{CO_2} are consistent with
modern or mid-Pliocene topography and geography (Fig. S7). For F_{vegice} , land surface and
vegetation changes considered here are a consequence of mid-Pliocene F_{CO_2} and F_{geotop} ⁴³, we
therefore separate F_{vegice} from the combined F_{CO_2} and F_{geotop} .

F_{vegice} explains ~78% (2.2 mm/day) of the regional mean $\delta(P-E)$ induced by F_{all} (2.8
167 mm/day) across the subtropical Sahel and East Asia (Fig. 4e and 4f), while contributions from
168 F_{CO_2} and F_{geotop} are small (Fig. 3a – d). Furthermore, only F_{vegice} produces a similar level of proxy-
169 model agreement in $\delta(P-E)$ compared to F_{all} . Both F_{CO_2} and combined F_{geotop} and F_{CO_2} produce low
values of Gwet's AC (Fig. 2b). The proxy-model agreement seen in the full forcing (F_{all})
experiment is only reproducible in experiment forced with F_{vegice} .

Compared to F_{all} , F_{vegice} accounts for ~50% of the increase in the global mean surface
temperature (2.7°C) and 58% of the increase (0.45 g/kg) in the global mean tropospheric (100 to
1000 hPa) specific humidity. In the Northern subtropics (between 20°N and 30°N), F_{vegice} explains
an even greater fraction of tropospheric moistening (61%, or 0.56 g/kg). In contrast, F_{CO_2} accounts
for 45% of the global mean surface warming (2.4°C), 31% (0.24 g/kg) of the global mean
tropospheric moistening, and only 26% of the tropospheric moistening in the northern subtropics.
F_{geotop} has much smaller influences on temperature and moisture responses globally. The influence
of F_{geotop} is slightly elevated in the northern subtropics accounting for 13% of the increase (0.11
180 g/kg) in the tropospheric humidity. Warming due to F_{vegice} is attributable to both lowered surface
albedo and enhanced evapotranspiration. Areas where boreal forest shifts and expands northward
and where mid-latitude deserts becomes vegetated feature substantially lowered surface albedo
(Fig. 3c). Expanded boreal forests also show large increases in upward latent heat flux, suggesting

Commented [FR17]: rewritten according to reviewer 1's suggestion

Commented [FR18]: "the" removed according to Reviewer 1

Commented [FR19]: "of the" added

Commented [FR20]: added "between...and..."

enhanced water vapor feedback⁴⁶ (Fig. 3b). A similar increase in latent heat flux also occurs in the
northern subtropics where $\delta(P-E)$ is positive.

Commented [FR21]: "A" added according to reviewer 1

Dynamical linkage between climate forcing conditions and mid-Pliocene hydroclimate

In order to identify the dynamical linkage between climate forcing conditions and
hydroclimate in the subtropical Sahel and East Asia, we adapt previously published moisture
budget diagnostics (MBD)⁴⁷ to decompose simulated June to September $\delta(P-E)$ by individual
models (Methods) into changes in the seasonal cycle of tropospheric moisture content ($\delta(P-E)_t$),
changes in the zonal mean circulation dynamics ($\delta(P-E)_{[V]}$) and stationary wave dynamics ($\delta(P-E)_{V^*}$),
changes in the tropospheric moisture content ($\delta(P-E)_Q$), changes in the interactions between
moisture and circulation dynamics ($\delta(P-E)_{VQ}$), and a residual term ($\delta(P-E)_{Resi}$) (Fig. 4 and Fig. S8).
The stationary wave response is quantified as the temporal mean departure from the zonal mean
following the classic circulation decomposition⁴⁸. The residual term combines the effects of
variability of transient eddies and changes of topography (see Methods).

As revealed by MBD, positive $\delta(P-E)$ in the subtropical Sahel and East Asia primarily
arises from changes in stationary wave dynamics ($\delta(P-E)_{V^*}$) and increased tropospheric moisture
content ($\delta(P-E)_Q$) (Fig. 4b and c). Contributions from both $\delta(P-E)_t$ and $\delta(P-E)_{VQ}$ are insignificant
in both regions (Fig. S8). Intermodel spread in simulated $\delta(P-E)$ strongly scales with the spread in
$\delta(P-E)_{V^*}$ and $\delta(P-E)_Q$, with little dependency on other terms (Fig. S10). Moreover, the MBD results
are consistent between the PlioMIP2 ensemble mean, and CESM2 experiments with F_{all} and F_{vegice}
(Fig. 4e and f, and Fig. S8). These results suggest a strong dynamical linkage between $\delta(P-E)_{V^*}$,
$\delta(P-E)_Q$, and $\delta(P-E)$ across PlioMIP2 models and that this dynamical linkage can mostly be
reproduced with F_{vegice} .

Commented [FR22]: added "the"

Commented [FR23]: changed to "between ... and ..."
according to reviewer 1

We further decompose the stationary wave response into components associated with
different wave numbers through Fourier decomposition of stream function anomalies (ψ_a) at 600
209 hPa. ψ_a is calculated as departures from the zonal mean and preindustrial (Fig. 3d - f). As shown
by the Fourier decomposition, zonal wave number 1 of ψ_a accounts for 67% of the total wave
energy between 0 - 90°N. This wave component features cyclones separately centered over
western Europe and North Africa, and anticyclones centered over the North Pacific (Fig. 3d - f).
Following the southern edge of the cyclonic wave centers, a moisture transport corridor emerges:
westerly winds bring moisture to North Africa from the tropical Atlantic; southerly and
southwesterly winds bring moisture from the tropical Indian and Pacific Oceans towards the
Indian subcontinent and East Asia. Furthermore, the rotational winds of this wave component are
connected to the surface warming pattern via the thermal wind relationship. This pattern is
broadly reproduced in the simulation forced with F_{vegice} (Fig. 3f).

Commented [FR24]: spelled out according to reviewer 2

Commented [FR25]: replaced "centered in" with
"centered over" according to the suggestion of reviewer 1

Positive $\delta(P-E)$ in both regions also results from increased tropospheric moisture content
($\delta(P-E)_Q$). Under PI conditions, low level winds converge towards North Africa and subtropical
East Asia, with diverging flow in the adjacent regions during the boreal summer (Fig. 4c). This
circulation is a key feature of the regional summer monsoon⁴⁹. Even without changing this
circulation, elevated tropospheric moisture content can result in greater moisture convergence and
positive $\delta(P-E)_Q$ in the subtropical Sahel and East Asia, and greater moisture divergence and
negative $\delta(P-E)_Q$ in the adjacent regions. This $\delta(P-E)_Q$ pattern is a known signature of
thermodynamic response to elevated CO_2 , i.e., the wet-gets-wetter, dry-gets-drier paradigm^{7,12,50}.
As shown in F_{vegice} , a similar thermodynamic response can be induced through land cover changes.

Commented [FR26]: added "the"

Discussion

**Comparison with previously proposed mechanisms**

The MBD reveals that both $\delta(P-E)_{[V]}$ and $\delta(P-E)_{Resi}$ are small (Fig. 4a and d). A small $\delta(P-$
$E)_{[V]}$ suggests little contribution from the zonal mean Hadley Circulation change to $\delta(P-E)$. If the
Hadley Circulation change were to drive boreal summer $\delta(P-E)$, coherent $\delta(P-E)$ across longitudes
between 0 and 30°N would be expected¹⁸. This is not the case as shown by the simulated $\delta(P-E)$
in ensemble mean (Fig. 1b) and individual experiments of PlioMIP2 (Fig. S6). Most PlioMIP2
simulations display a clear latitudinal offset in positive $\delta(P-E)$ between the East Asia and west
Pacific (Fig. S6) and between the North Africa and central America.

A small $\delta(P-E)_{Resi}$ suggests a small net influence from changes of transient eddies and
topography on $\delta(P-E)$ (Fig. 4d). Tropical SST warming during the El Niño events influences
extratropical precipitation by strengthening and shifting the positions of storm track and the
subtropical jet equatorward²⁵. These changes are not observed in mid-Pliocene simulations of the
northern extratropics (Fig. S9). In this area, storm activity weakens as shown by changes in the
850 hPa eddy kinetic energy (EKE). The subtropical jet, measured as the 200 hPa zonal wind
speed, across Eurasia also weakens but remains in a similar position to pre-industrial. These
changes are consistent with a reduction in the poleward temperature gradient⁵¹ (Fig. S9a and b).
Moreover, the pattern of precipitation variability induced by El Niño SSTs⁵² has a classic signature
of negative $\delta(P-E)$ in North Africa paired with positive $\delta(P-E)$ in Southeast Asia. This signature is
not observed in mid-Pliocene simulations: $\delta(P-E)$ is broadly consistent across both regions (Fig.
1b and S6). Models suggest that El Niño-like SSTs therefore do not play a significant role driving
mid-Pliocene $\delta(P-E)$ in these regions.

However, $\delta(P-E)_{Resi}$ is more noticeable in the North Pacific and subtropical western North
American. The impact of El Niño-like SSTs could be important for $\delta(P-E)$ in both regions¹³. Past
hydroclimate changes in the western North America have been shown to be sensitive to changes
in eddy moisture transport associated with varying conditions of atmospheric rivers^{53,54,55}. $\delta(P-E)$
changes in this region are likely driven by processes different from the subtropical Sahel and East
Asia.

**Implications for the hydrological cycle during Cenozoic warm intervals**

The stationary wave response identified here is distinct from a zonal mean Hadley
Circulation response or ITCZ shift. These mechanisms highlight the importance of a changing
zonal mean energy budget on circulation dynamics^{14,18,56-60}. Instead, the stationary wave response
in PlioMIP2 experiments is generated by a spatially heterogeneous warming pattern induced by
continental greening. The circulation change in the free lower troposphere can be explained by this
surface warming pattern through the diagnostic thermal winds, which closely follow the zonal
wave number 1 of ψ_a . This “pattern effect” has previously been overlooked when examining past
hydroclimate changes.

Why is F_{vegice} more effective at altering subtropical terrestrial hydroclimate compared to
F_{CO2} ? Land cover changes are known to be key to generating local surface temperature and regional
hydroclimate responses across North Africa and subtropical East Asia⁶¹⁻⁶³ and may even alter the
strength of the Atlantic meridional overturning circulation^{64,65}. In North Africa, expansion of
desert results in enhanced surface albedo, surface cooling, and strengthened diabatic subsidence
and dust emission^{66,67}. These responses may further suppress moist convection and perpetuate
desert⁶⁸⁻⁷¹. A positive vegetation-precipitation feedback results in multiple equilibrium states of
North African vegetation over the late Quaternary⁶². In our simulations, atmospheric circulation
responses to F_{vegice} facilitate moisture transport towards the subtropical Sahel and East Asia via a
stationary wave response. This response is closely tied to the large-scale warming pattern

Commented [FR27]: added “the” according to reviewer 1

Commented [FR28]: Text separated from the “Results” section to form a separate discussion on the influence of El Niño-Like SST and Hadley circulation on the subtropical Sahel and East Asia hydroclimate – in response to Reviewer 1’s comment
Lines 205 to 221 in the previous version

Commented [FR29]: reference added following the suggestion of reviewer 1

generated by F_{vegice} , which does not occur in F_{CO_2} (Fig. S11). Additionally, this response also
differs from the regional wave response driven by diabatic heating from the South Asian monsoon
moist convection as suggested by previous studies⁶³. The latter features contrasting wave center
and hydroclimate responses between South Asia and western Sahel, distinct from our findings of
a continental-scale wave pattern with consistent responses between these two regions.

Different hydroclimate responses to F_{CO_2} and F_{vegice} also reflect the difference between
future and mid-Pliocene land surface processes. Increasing CO_2 may favor reduction of soil
moisture and partitioning of surface heat flux towards sensible heat, leading to enhanced surface
warming. This in turn lowers relative humidity above the surface and diminishes continental cloud
cover, contributing to negative $\delta(\text{P-E})$ on land. A similar response of surface heat flux partitioning
and moisture deficit can be caused by the reduction of leaf transpiration in response to CO_2
fertilization. Soil moisture feedbacks and CO_2 fertilization drive much of the predicted future
subtropical terrestrial hydroclimate change¹⁵. These changes may also contribute to the muted $\delta(\text{P-E})$
response to F_{CO_2} in our experiments despite a modest increase in precipitation. In contrast, F_{vegice}
features no physiological effect of CO_2 and produces large increases in latent heat flux relative to
sensible heat flux across continents (Fig. S12), creating favorable conditions for a more humid
troposphere. Humidification of the subtropical Sahel and East Asia therefore reflect a synergy of
dynamic and thermodynamic responses to F_{vegice} . Reduced ice sheet cover, expansive northern
high-latitude boreal forests, and vegetated Sahel and Central Asia have all been recorded during
other Cenozoic warm intervals⁷²⁻⁷⁴. These land cover changes were likely instrumental in driving
global mean temperature and the terrestrial hydrological cycle throughout the Cenozoic.

Past hydroclimate states driven by Earth System Feedbacks

Our results suggest an alternate view of the role of vegetation changes in modulating
continental hydroclimate. Changes in regional circulation in the form of stationary wave responses
lead to strengthened low-level winds that import moisture into the subtropical Sahel and East Asia
from the tropical Atlantic and Indian Oceans, respectively. This process is amplified by enhanced
tropospheric moistening. This mechanism is distinct from previous mechanisms that highlight the
role of the zonal mean Hadley Circulation or local ecosystem responses to land surface changes.
Although complete feedbacks from vegetation and ice sheets are not prognostically simulated in
most mid-Pliocene simulations (Supplementary Table 2), the presented simulations highlight that
radiative perturbations from proxy-constrained changes in vegetation and ice sheets are key to
generating hydroclimate changes in the northern subtropics. Moreover, these hydroclimate
changes have minimal dependency on the prescribed mid-Pliocene topography and land-sea
distribution (Fig. 1g).

A key implication of our findings is that the mid-Pliocene continental hydroclimate is more
appropriately viewed as part of the Earth system feedbacks instead of an immediate response to
F_{CO_2} . This inference is supported by the strong relationship between a model's ability to capture
the mid-Pliocene hydroclimate state and its simulated global mean warming (Figure 4). The latter
reflects model diversity in Earth system sensitivities to the F_{CO_2} as well as the vegetation and ice
sheet conditions of the mid-Pliocene. In contrast, the relationship between a model's ability to
capture the mid-Pliocene hydroclimate state and its equilibrium climate sensitivity is weak.
Equilibrium climate sensitivities of individual models are estimated from doubling CO_2
experiments with the same model and resolution³³. Published studies have mostly focused on the
impact of Earth system feedbacks on temperature responses^{43,75}. Here, we demonstrate that these

Commented [FR30]: Rephrased in response to reviewer 1's comment

Commented [FR31]: added "the"

Commented [FR32]: added "the"

feedbacks are key for understanding paleohydrological responses. Inconsistencies in
interpretations of proxy hydroclimate records^{18,76} may be resolved by considering F_{vegice} .

Lastly, our results offer a resolution to the apparent discrepancies between the projections
of hydroclimate changes in the Sahel and subtropical East Asia following the middle-of-the-road
scenarios and strong geologic evidence for moist Pliocene climate at similar levels of CO₂. While
the near-future is dominated by the short-term response to CO₂ radiative forcing and internal
variability, Pliocene hydroclimate reflects long-term adjustments of the Earth system that
incorporates responses from vegetation and ice sheets, which occur on timescales longer than most
future climate projections. Feedbacks of vegetation and ice sheets to CO₂ increase are known to
amplify the response of equilibrium surface temperature to radiative forcing^{43,75}. We highlight that
these relatively slow Earth system feedbacks are also critical for understanding Earth's
hydroclimate responses to varying CO₂. Therefore, changes in vegetation and ice sheet
distributions should be carefully considered when simulating past and future climate.

**Materials and Methods**

**Proxy-Model Comparison**

Our proxy compilation builds on previous efforts to compile Pliocene hydroclimate
records. These include the compilations created by Refs^{18,40,41}. The records included in these
sources span a variety of proxy types, from sedimentological indicators of lacustrine
environments, palynological indicators of vegetation composition, faunal remains, and stable
isotope records from organic and inorganic materials. We added to these compilations by
including new records of terrestrial hydroclimate dating to the Pliocene archived on the NOAA
Paleoclimatology Database (<https://www.ncdc.noaa.gov/data-access/paleoclimatology-data>) and
Pangaea Database (<https://www.pangaea.de/>).

To identify the average hydroclimate signal during the mid-Pliocene, we filtered
available records based on the precision of their age models. Specifically, proxy records were
required to include at least two age control points, with one age control point after the mid-
Pliocene and a clearly identifiable age control point prior to mid-Piacenzian. For many records,
this 'basal' tie point was often in the early Pliocene or Miocene. Only including records with
multiple age control points reduces the likelihood that samples inferred to be from the mid-
Pliocene actually come from earlier in the Pliocene or the early Pleistocene. For most proxy
records, age control derives from a combination of magnetostratigraphy (e.g. the Gauss-
Matuyama boundary), radiometric dating, or biostratigraphic information. In some cases,
identification of independently-dated tephra layers or correlation with the benthic oxygen isotope
stack provides age control. These filtering criteria allowed us to retain a compilation of 62
records. Of the 62 records included, 30 come from paleobiological indicators like faunal remains
or pollen, while the other 34 records are drawn from interpretations of sedimentary sequences or
stable isotopic analyses of organic and inorganic materials (SI). A supplementary excel file (Data
S1) with details on the proxy records, including methods, chronology, and original citation, is
included with this manuscript. Because of the broad, continental-scale coherence of the model
signal, proxy-model synthesis results are similar when using the compilations found in other
sources e.g. Ref⁴¹.

We rely on the author's original interpretation about whether the record reflects, on
average, wetter or drier conditions, or no change in hydroclimate during the mid-Pliocene
compared to late Quaternary/modern conditions. Many records are discontinuous, and cannot
provide quantitative comparisons between mid-Pliocene climate and late Quaternary or pre-

Commented [FR33]: rephrased, sentence commented by reviewer 1

Commented [FR34]: added "the" according to reviewer 1

[revised manuscript text omitted]

 $R(F_{\text{all}}) = R(F_{\text{vegice}}) + R(F_{\text{geotop}}) + R(\text{CO}_2)$, for which $R(F_{\text{all}}) = R(\text{Eoi400}) - R(\text{E280})$; $R(F_{\text{geotop}}) =$
 $R(\text{Eo400}) - R(\text{E400})$ or $R(F_{\text{geotop}}) = R(\text{Eo280}) - R(\text{E280})$; $R(F_{\text{CO}_2}) = R(\text{E400}) - R(\text{E280})$ or
 $R(F_{\text{CO}_2}) = R(\text{Eo400}) - R(\text{Eo280})$; $R(F_{\text{vegice}}) = R(\text{Eoi400}) - R(\text{Eo400})$.

Development of moisture budget analysis

To diagnose the causes of $\delta(P-E)$ in model simulations, we further developed and applied
 the moisture budget analysis⁴⁷ to the multimodel mean of the PlioMIP2 simulations. With only
 monthly data available, our derivation aims to facilitate the comparison of a pair of experiments
 and the evaluation of existing hypotheses regarding the cause of mid-Pliocene hydroclimate
 change (e.g. identifying whether mid-Pliocene hydroclimate anomalies results from zonal mean
 changes or stationary wave changes).

Following equation (13) in Ref⁴⁷ on pressure coordinates, precipitation minus evaporation
 is balanced by changes in the moisture tendency and moisture convergence :

$$494 \quad P - E = -\frac{1}{g\rho_w} \frac{\partial}{\partial t} \int_0^{P_s} q dp - \frac{1}{g\rho_w} \nabla \cdot \int_0^{P_s} u q dp \quad (1)$$

In Equation (1), g is geopotential acceleration, ρ_w is the density of water, P_s is surface pressure,
 and q is specific humidity, p : pressure. For a pair of experiments, experiment 1 is the control case
 (e.g. preindustrial) and experiment 2 is the sensitivity experiment (e.g. Pliocene), the small
 perturbation method tells us that $q_2 = q_1 + \delta q$; $u_2 = u_1 + \delta u$; $(P - E)_2 = (P - E)_1 +$
 $\delta(P - E)$; and $P_{s2} = P_{s1} + \delta P_s$. We can therefore decompose the perturbations to P-E into
 contributions from the anomalous moisture tendency, changes in convergence due to changes in
 wind, moisture, and the interaction of these changes, as well as a residual term:

$$\delta(P - E) = -\frac{1}{g\rho_w} \frac{\partial}{\partial t} \int_0^{P_{s1}} \delta q dp - \frac{1}{g\rho_w} \nabla \cdot \int_0^{P_{s1}} (q_1 \delta u + u_1 \delta q + \delta u \delta q) dp + resi1$$

$$(2)$$

$$resi1 = -\frac{1}{g\rho_w} \frac{\partial}{\partial t} q_2 \delta P_s - \frac{1}{g\rho_w} \nabla \cdot (u_2 q_2 \delta P_s) \quad (3)$$

Residual term 1 (*resi1*) quantifies the results from changing surface pressure, which is
 dominated by topographic changes between the mid-Pliocene and pre-industrial. Applying
 Reynold's decomposition, we can separate the total change in a given quantity into its temporal
 mean (e.g. monthly or annual climatology), denoted via an overbar, and higher frequency temporal
 fluctuations, denoted via a prime. Such that:

$$\delta q = \overline{\delta q} + \delta q', \delta u = \overline{\delta u} + \delta u' \quad (4)$$

$$\delta(P - E) = \overline{\delta(P - E)} + \delta(P - E)', u_1 = \overline{u_1} + u_1', q_1 = \overline{q_1} + q_1', P_{s1} = \overline{P_{s1}} + P_{s1}' \quad (5)$$

Given that we are interested in understanding the contributions from zonal mean circulation
 changes compared to other changes, we further separate δu into changes in zonal mean,
 indicated by square brackets ($[\]$) and changes in deviations from the zonal mean (i.e., the
 stationary wave), which is indicated by an asterisk (*):

$$\delta u = [\delta u] + \delta u^*$$

Using these expansions, we can rephrase the anomalous moisture budget in equation (2) to separate
 out the contributions due to zonal mean and stationary wave. The updated form of equation 2 is:

$$\overline{\delta(P - E)} = -\frac{1}{g\rho_w} \frac{\partial}{\partial t} \int_0^{\overline{P_{s1}}} \overline{\delta q} dp - \frac{1}{g\rho_w} \nabla \cdot \int_0^{\overline{P_{s1}}} (\overline{u_1} \overline{\delta q} + \overline{q_1} ([\delta u] + \delta u^*) + \delta u \delta q) dp + resi1 +$$

$$resi2 \quad (6)$$

Residual term 2 (*resi2*) quantifies combined effects of transient eddies moisture transport and
 influxes:

$$resi2 = -\frac{1}{g\rho_w} \frac{\partial}{\partial t} \overline{\delta q' P_s'} + \frac{1}{g\rho_w} \nabla \cdot [(\overline{u_1} \delta q' + u_1' \delta q + q_1' \delta u + \overline{q_1} \delta u' + \delta u' \delta q + \delta u \delta q') P_s']$$

$$- \frac{1}{g\rho_w} \nabla \cdot \int_0^{\overline{P_{s1}}} (\delta q' \delta u' + q_1' \delta u' + u_1' \delta q') dp$$

Calculating residual terms 1 and 2 require high frequency outputs of surface pressure, three-
 dimensional specific humidity and horizontal winds, which are not available. Also, even at 6-
 hourly resolution, the calculated eddy terms are insufficient to close the moisture budget with
 reanalysis observational data⁴⁷. Thereby, *resi1* and *resi2* are not explicitly calculated in our
 calculations and quantified as a combined residual (i.e., the difference between P-E changes and
 the sum of other terms in the moisture budget equation). Yet, the decomposition demonstrates
 physical interpretations of these residuals. Monthly climatologies of surface pressure,
 precipitation, evaporation, and three-dimensional horizontal winds, and specific humidity are used
 to calculate the remaining terms in equation 6. From the left to right, the first term in equation 6
 $(-\frac{1}{g\rho_w} \frac{\partial}{\partial t} \int_0^{\overline{P_{s1}}} \overline{\delta q} dp)$ is the moisture tendency term, which quantifies contributions to $\overline{\delta(P - E)}$

from changes in seasonal cycle. The term $-\frac{1}{g\rho_w}\nabla\cdot\int_0^{P_{s1}}(u_1\delta\bar{q})dp$ describes contributions from
changes in climatological mean tropospheric moisture content; $-\frac{1}{g\rho_w}\nabla\cdot\int_0^{P_{s1}}(q_1[\delta u])dp$ describes
contributions from changes in zonal mean circulation; $-\frac{1}{g\rho_w}\nabla\cdot\int_0^{P_{s1}}(q_1\delta u^*)dp$ describes
contributions from changes in stationary wave kinetics. Finally, $-\frac{1}{g\rho_w}\nabla\cdot\int_0^{P_{s1}}(\delta u\delta q)dp$ describes
contributions from covarying changes in mean moisture content and horizontal circulation.

**Acknowledgments**

The authors would like to thank all the modelling groups who provided the PMIP3 and
PMIP4 outputs for this analysis, WCRP, CMIP panel, PCMDI, ESGF infrastructures for
sharing data, WCRP and CLIVAR for supporting the PMIP project.

**Computing:** The CESM2 simulations are performed with high-performance computing
support from Cheyenne (doi:10.5065/D6RX99HX) provided by NCAR's Computational
and Information Systems Laboratory, sponsored by the National Science Foundation.

The IPSL-CM6A-LR simulation was run on the Très Grande Infrastructure de Calcul
(TGCC) at Commissariat à l'Energie Atomique (gencmip6 project) under the allocations
2016-A0030107732, 2017-R0040110492 and 2018-R0040110492 (project gencmip6)
provided by GENCI (Grand Equipement National de Calcul Intensif).

The model simulations with EC-Earth3 and the data analysis were performed using
resources provided by ECMWF's computing and the Swedish National Infrastructure for
Computing (SNIC) at the National Supercomputer Centre (NSC), which is partially
funded by the Swedish Research Council through grant agreement no. 2018-05973.

AAO and WLC acknowledge JAMSTEC for use of the Earth Simulator supercomputer.

**Funding:**

U.S. National Science Foundation grant 1814029 (R. Feng)

U.S. National Science Foundation grant 1903650 (R. Feng)

U.S. National Science Foundation grant 1903148 (T. Bhattacharya)

France ANR HADoC (ANR-17-CE31-0010) (C. Contoux).

Swedish Research Council (Vetenskapsrådet; grant nos. 2013-06476 and 2017-04232)

(Q. Zhang)

Japanese JSPS Kakenhi grant 17H06104 and MEXT Kakenhi grant 17H06323 (W.-L.

Chan and A. Abe-Ouchi)

The PRISM4 reconstruction and boundary conditions used in PlioMIP2 were funded by
the U.S. Geological Survey Climate and Land Use Change Research and Development Program.

Any use of trade, firm, or product names is for descriptive purposes only and does not imply

endorsement by the U.S. Government.

**Author contributions:** Conceptualization: RF, TB, BOB, AMH

Methodology: RF, TB, MC

Investigation: RF, TB

Visualization: RF, TB

Writing—original draft: RF, TB

Writing—review & editing: all authors

**Competing interests:** none.

**Data availability:** PlioMIP2 simulations that are part of the Climate Model Intercomparison
Project 6 (CMIP6) can be found via Earth System Grid Federation:

EC-Earth3: <https://doi.org/10.22033/ESGF/CMIP6.4804>

IPSL-CM6A-LR: <https://doi.org/10.22033/ESGF/CMIP6.5230>

GISS-E2-1-G: <https://doi.org/10.22033/ESGF/CMIP6.7227>

CESM2: <https://doi.org/10.22033/ESGF/CMIP6.7675>

HadGEM3: <http://doi.org/10.22033/ESGF/CMIP6.12130>

PRISM4 boundary condition data sets can be found here:

https://geology.er.usgs.gov/egpsc/prism/7.2_pliomip2_data.html

Source data from PlioMIP2 ensemble and CESM2 are provided with this paper at Zenodo:

<https://zenodo.org/record/5706370#.YZbc6y1h0eY>

**Code availability:**

Source code of CESM2 can be downloaded from

https://escomp.github.io/CESM/versions/cesm2.1/html/downloading_cesm.html.

The code for moisture budget analysis is published at Zenodo:

<https://zenodo.org/record/5706370#.YZbc6y1h0eY>

Commented [FR35]: data now published at Zenodo

 **Fig. 1. $\delta(P-E)$ in Pliocene proxy records and both future and Pliocene simulations.** (a) $\delta(P-E)$
 between 2081 to 2100 and 1986 to 2005 following Shared Socioeconomic Pathway 2-4.5
 simulated by models participating in Climate Model Intercomparison Project 6. (b) – (c) Pliocene
 annual and boreal summer (June to September) mean $\delta(P-E)$ of PlioMIP2 ensemble. Proxy data
 displayed as circles. (d) Correspondence between the subtropical Sahel ($10^{\circ} - 20^{\circ}N$, $10^{\circ}W - 25^{\circ}E$) and East Asia ($20^{\circ} - 30^{\circ}N$, $80^{\circ}E - 100^{\circ}E$) $\delta(P-E)$ simulated by PlioMIP2 experiments and
 a subset of SSP245 experiments with the same models (model names are marked with asterisks).
 (e) to (h) $\delta(P-E)$ in response to full Pliocene climate forcing conditions (F_{all}), CO_2 forcing alone
 (F_{CO_2}), changes in geography and topography (F_{geotop}), and changes in vegetation and icesheet
 (F_{vegice}) simulated by Community Earth System Model version 2. SSP2-4.5 ensemble includes

Commented [FR36]: added “.” according to both reviewers

Commented [FR37]: replotted in response to reviewer 2

Commented [FR38]: rephrased according to reviewer 1's suggestion

Commented [FR39]: corrected for the missing “.”

BCC-CSM2-MR, CESM2, CESM2-WACCM, CanESM5, CNRM-CM6-1, CNRM-ESM2-1,
 EC-Earth3.3, GISS-E2-1-G, HadGEM3-GC31-LL, IPSL-CM6A-LR, MIROC6, MIROC-ES2L,
 MRI-ESM2-0, NESM3, UKESM1-0-LL. In (b) to (h), $\delta(P-E)$ is the difference between the
 Pliocene and preindustrial simulations. Area significant against multi-model spread is hatched in
 (a) to (c) identified by Welch's t-test ($p < 0.1$).

**Fig. 2. Degree of agreement between proxy hydroclimate indicators and simulated $\delta(P-E)$,**
 **and correlation between annual mean and boreal summer signal.** Proxy-model fit is assessed
 using a measure of categorical agreement between two datasets called Gwet's AC. For a given %
 threshold change in P-E in models, higher (lower) values indicate that proxies and models agree
 (disagree) that a given location is wet, dry, or neutral. Area within dashed line indicates
 statistically significant agreements. a) Gwet's AC agreement between individual models, MMM,
 and proxies at different thresholds. CMIP6 models are identified with an asterisk. b) Agreement
 between proxies and $\delta(P-E)$ in response to F_{CO_2} , combined F_{CO_2} and F_{geotop} , F_{vegice} and F_{all}
 simulated by CESM2. c) Gwet's AC agreement between annual and boreal summer averages at
 proxy sites for PlioMIP2 ensemble mean. CMIP6 models are identified with an asterisk.

Commented [FR40]: added in response to reviewer 2

Commented [FR41]: corrected according to reviewer 1's comment

 **Fig. 3. Pliocene vegetation and ice sheet distribution, and responses of boreal summer**
 **surface radiation, temperature, and stationary wave to F_{all} and F_{vegice} .** (a) vegetation and ice
 ice sheet distribution: the plant functional type (PFT) with the highest percentage in a grid cell is
 shown. (b) Surface latent heat flux and (c) albedo changes induced by F_{vegice} . (d) – (f) changes of
 surface temperature (color shaded), and 600 hPa wave-number 1 (contour) and rotational winds
 (vectors) of stationary wave in response to F_{all} simulated by PlioMIP2 ensemble and CEM2,
 and to F_{vegice} simulated by CEM2.

 **Fig. 4. Moisture budget decomposition of boreal summer $\delta(P-E)$ and the potential linkage**
 **of $\delta(P-E)$ with Earth System Sensitivity (ESS) and Equilibrium Climate Sensitivity (ECS).**
 (a) to (d) contributions to $\delta(P-E)$ from zonal mean circulation ($\delta(P-E)_{[V]}$), changes of stationary
 wave dynamics ($\delta(P-E)_{[V^*]}$), tropospheric moistening ($\delta(P-E)_O$), and a residual term ($\delta(P-E)_{[res]}$).
 (e) and (f) Regional mean (red boxes in (a) to (d)) moisture budget for the subtropical Sahel and
 East Asia simulated by PlioMIP2 ensemble, and CESM2 with F_{all} and F_{vegice} . Error bars show 1
 standard deviations of model spread. (g) to (h) Gwet's ACs of individual PlioMIP2 simulations as
 a function of global mean warming (δTs , which quantifies ESSs), and ECSs of these models. δTs
 and ECSs are from Ref³³. Boxplots show the spread of Gwet's AC from using 20% to 60% of
 $\delta(P-E)$ as thresholds (Method).

Commented [FR42]: caption corrected in response to reviewer 1

[revised manuscript text omitted]

- 84. Chan, W.-L. & Abe-Ouchi, A. PlioMIP2 simulations using the MIROC4m climate model.
*Climate of the Past* 1–35 (2020).
- 85. Tan, N. *et al.* Modeling a modern-like pCO₂ warm period (Marine Isotope Stage KM5c)
with two versions of an Institut Pierre Simon Laplace atmosphere–ocean coupled general
circulation model. *Climate of the Past* **16**, 1–16 (2020).
- 86. Li, X., Guo, C., Zhang, Z., Otterå, O. H. & Zhang, R. PlioMIP2 simulations with
NorESM-L and NorESM1-F. *Climate of the Past* **16**, 183–197 (2020).
- 87. Williams, C. J. *et al.* Simulation of the mid-Pliocene Warm Period using HadGEM3:
Experimental design and results from model-model and model-data comparison. *Climate*
*of the Past* 1–45 (2021).
- 88. Brierley, C. M. & Fedorov, A. V. Comparing the impacts of Miocene–Pliocene changes in
inter-ocean gateways on climate: Central American Seaway, Bering Strait, and Indonesia.
*Earth and Planetary Science Letters* **444**, 116–130 (2016).
- 89. Otto-Bliesner, B. L. *et al.* Amplified North Atlantic warming in the late Pliocene by
changes in Arctic gateways. *Geophysical Research Letters* **44**, 957–964 (2017).
- 90. Hill, D. J. The non-analogue nature of Pliocene temperature gradients. *Earth and*
*Planetary Science Letters* (2015).

Supplementary Materials for

Past terrestrial hydroclimate sensitivity controlled by Earth System Feedbacks

R. Feng^{1*}, T. Bhattacharya², B. Otto-Bliesner³, E. Brady³, A. Haywood⁴, J. Tindall⁴, S. Hunter⁴, A. Abe-Ouchi⁵, W. -L. Chan⁵, M. Kageyama⁶, C. Contoux⁶, C. Guo⁷, X. Li⁸, G. Lohmann⁹, C. Stepanek⁹, N. Tan¹⁰, Q. Zhang¹¹, Z. Zhang⁸, Z. Han¹², C. J. R. Williams¹³, D. J. Lunt¹³, H. Dowsett¹⁴, D. Chandan¹⁵, W. R. Peltier¹⁵

*Corresponding author. Email: ran.feng@uconn.edu

This PDF file includes:

Table S1

Table S2

Figs. S1 to S13

References (80 to 113) (if applicable—these should refer only to references in the SM)

Other Supplementary Materials for this manuscript include the following:

Data S1

**Supplementary Text**

Sensitivity experiments using CESM2

We carried out two new simulations using the community Earth System Model version 2
 (individual model components are Community Atmospheric Model version 6, Community Land
 Model version 5, Parallel Ocean Program version 2, and Community Ice CodE version 5) to
 quantify individual effects of elevated CO₂, changes in paleo-geography and topography, and
 vegetation and ice sheets on mid-Pliocene P-E changes. These new experiments separately
 feature Pliocene levels of carbon dioxide (400 ppm CO₂) coupled with preindustrial boundary
 conditions (e.g. ice sheets and vegetation and geography and topography) in the case of E400.
 This simulation was used to isolate the influence of Pliocene CO₂ (F_{CO2}). For Eo400, the
 simulation features preindustrial vegetation and ice sheets and otherwise mid-Pliocene CO₂ and
 boundary conditions, and can be compared to the full Pliocene simulation to isolate the influence
 of vegetation and ice cover (F_{vegice}). The influence of geography and topography (F_{geotop}) was
 isolated by subtracting E400 from Eo400.

 Eo400 and E400 are initialized with ocean states and terrestrial carbon and nitrogen states from
 previously published runs of the mid-Pliocene, and preindustrial which feature the same
 geography and topography as Eo400 and E400 respectively. Each of which was run for more
 than 500 model years. Model equilibrium is diagnosed with global mean net top of atmosphere
 radiation imbalance (F_{net}) and global mean surface temperature (T_s). For the last 100 model
 50 years, global mean F_{net} of all simulations is ~ 0.2W/m² for both simulations, and trends of global
 mean T_s are 0.1 and 0.2°C per century (Fig. S9).

 Proxy recorded Pliocene regional hydroclimate patterns

We compiled Pliocene hydroclimate indicators from published records (Data S1). We rely on the
 author's original interpretations about whether the record reflects, on average, wetter or drier
 conditions, or no change in hydroclimate during the mid-Pliocene compared to late
 Quaternary/modern conditions. Below, we provide a review of published studies on broad
 regional trends evident in the proxy records.

**Supplemental Table 1** Broad regional trends of mid-Pliocene hydroclimate changes recorded by
 a compilation of proxy data.

Regions	Summary of published mid-Pliocene hydroclimate changes
Europe and the Mediterranean	Evidence of wetter mid-Pliocene conditions in Europe and the Mediterranean primarily come from sedimentological evidence of expanded lacustrine environments and palynological indicators of more mesic vegetation. For instance, pollen and macrobotanical remains from the lagerstatte deposit at Willershausen in modern Germany provide evidence of a slightly wetter climate ¹ and pollen records ² document mesic vegetation across the Iberian Peninsula and north Africa. Evidence from the Dacian Basin records an interval of high salinity at 3 Ma towards the end of the mid-Pliocene, which could reflect changes to the regional water budget, but has been interpreted to reflect higher water levels in the Black Sea that slightly postdate the mid-Pliocene ³ . This coheres with the

Commented [FR1]: The text in the original draft is organized into a table.

	interpretations of ⁴ , which suggest a decrease in winter precipitation in midlatitude Europe between 4 and 3 Ma.
Africa and the Middle East	Continuous Plio-Pleistocene records of dust flux from off the coast of west Africa have been interpreted as evidence of long-term drying ^{5,6} . Several records also provide evidence of the expansion of ecosystems dominated by C4 plant species ⁷⁻¹⁰ . Like pollen records, the interpretation of these records is complex since these ecosystem shifts may reflect hydroclimate, but these may also be driven by changes in $p\text{CO}_2$ or fire regimes. However, records of stable isotopes of oxygen and hydrogen in organic and inorganic materials, which have been interpreted as reflecting an ‘amount effect,’ whereby higher rainfall rates result in a more depleted isotopic signature, reflect a wetter mid-Pliocene ^{7,11,12} . Pollen data, dust flux records, and sedimentological indicators also show wetter conditions in the Levant and Arabian Peninsula during the mid-Pliocene ^{13,14} . This is corroborated by a recently published stable isotopic record of hydroclimate from a cave in the Negev Desert ¹⁵ .
South and East Asia	Evidence of wetter mid-Pliocene conditions in South and East Asia are primarily drawn from palynological transfer functions or faunal remains ¹⁶⁻¹⁸ . However, in many regions these inferences are corroborated by other indicators ¹⁹ . In the Qaidam Basin (northeastern Tibetan plateau), and in southwest China’s Yuanmou region, sedimentological indicators provide evidence of a wetter mid-Pliocene in regions where pollen evidence suggests little change ^{14,20-23} . In the Loess Plateau region, multiple proxies provide evidence of no change or drier conditions at the mid-Pliocene compared to the Pleistocene ^{24,25} . However, evidence from mapping the total extent of loess deposits at different intervals during the Neogene and Quaternary provides evidence of a more mesic climate during the mid-Pliocene ²⁶ .

**Supplemental Table 2** Models, references, and vegetation boundary conditions of PlioMIP2
 simulations analyzed in this study.

	Modeling Group	Model reference	Vegetation
CCSM4	National Center for Atmospheric Research	²⁷	Prescribed according to PRISM4 ²⁸
CESM1.2	National Center for Atmospheric Research	²⁹	Prescribed according to PRISM4 ²⁸
CESM2	National Center for Atmospheric Research	³⁰	Prescribed according to PRISM4 ²⁸
COSMOS	Alfred Wegener Institute, Germany	³¹	Dynamic
EC-Earth 3.3	Stockholm University, Sweden	³²	Prescribed according to PRISM4 ²⁸
HadCM3	University of Leeds, UK	³³	Prescribed according to PRISM4 ²⁸
IPSL-CM5	Laboratoire des Sciences du Climat et de l'Environnement (LSCE), France	³⁴	Prescribed according to PRISM4 ²⁸
IPSL-CM5A2	Laboratoire des Sciences du Climat et de l'Environnement (LSCE), France	³⁴	Prescribed according to PRISM4 ²⁸
IPSL-CM6	Laboratoire des Sciences du Climat et de l'Environnement (LSCE), France	³⁵	Prescribed according to PRISM4 ²⁸
MIROC4m	Center for Climate System Research (Uni. Tokyo), JAMSTEC	³⁶	Prescribed according to PRISM4 ²⁸
NorESM-1L	Norwegian Research Centre, Bjerknes Centre	³⁷	Prescribed according to PRISM4 ²⁸
HadGEM3	UK Met Office	³⁸	Prescribed according to PRISM4 ²⁸
GISS-E2-1G	NASA Goddard Institute for Space Studies	³⁹	Prescribed according to PRISM4 ²⁸

**Fig. S1. $\delta(P-E)$ between mean (P-E) of 2081 to 2100 following Shared Socioeconomic**
**Pathway 2-45 and year 1850 preindustrial control (20-yr average) simulated by the same**
**set of CMIP6 models shown in Fig. 1. Hatched areas show statistically significant**
**differences identified by Welch's t-test ($p < 0.1$).**

**Fig. S2. Simulated terrestrial hydroclimate change between PlioMIP2 simulations and PI,**
 **referred to as $\delta(P-E)$, during the boreal winter (December to March).**

Fig. S3. Locations of sites used in our proxy compilation, set against a background of
 annual average rainfall rate from the GPCPC. Co-located sites were combined if they were
 less than 150 km apart and featured same-signed anomalies. These include: Site 35, 72; 77,
 22; 79,61; 65, 66; and 61,79,58,191. Site numbers correspond to site indices in supplemental
 file (SI_pliocene_hydroclimate.xlsx), which also contains information on proxies,
 chronology, and references. Note that numbers are non-continuous, since some sites in our
 original compilation were excluded by our quality control standards for proxy data.
 Original references for each site are included in the main text Methods section.

 **Fig. S4** Agreement between proxies and models, including the multi-model mean (MMM), in
 different seasons. (a) shows Gwet's AC value at different thresholds of % change in P-E for
 winter (December-March) rainfall only, and b) shows Gwet's AC values for summer rainfall only
 (July-September).

 **Fig. S5 Simulated changes in net primary productivity between mid-Pliocene and**
 **preindustrial by (a) Community Climate Model version 4 (CCSM4), (b) Community Earth**
 **System Model version 1 (CESM1) and (c) version 2 (CESM2).**

**Fig. S6. (a) Change in annual mean precipitation minus evaporation ($\delta(P-E)$) in individual**
 **PlioMIP2 models relative to preindustrial. Model name is given in the in top left of each**
 **panel, and proxy records are overlaid to show whether a given record shows wetter, drier,**
 **or no change. (b) Range of significance thresholds of $\% \delta(P-E)$ at which the agreement**
 **between proxies and models shown in Figure 2 in the main text is significant, with gray**
 **colors indicating significant values of Gwet's AC at that percentage threshold.**

 **Fig. S7 Annual mean $\delta(P-E)$ responses to F_{CO_2} with a) mid-Pliocene and b) PI geography**
 **and topography, and $\delta(P-E)$ responses to F_{geotop} with a) mid-Pliocene and b) PI CO_2 .**

 **Fig. S8** June to September contributions to $\delta(P-E)$ from (a) and (b) change in the seasonal cycle
 ($\delta(P-E)_t$), (c) and (d) stationary wave dynamics ($\delta(P-E)_{v^*}$), e) and f) tropospheric moistening
 ($\delta(P-E)_Q$), (g) and (h) residual combining the effect of transient eddies and topographic changes
 ($\delta(P-E)_{res}$), (i) and (j) covarying humidity and winds ($\delta(P-E)_{ov}$). Left column: $\delta(P-E)$ due to
 Pliocene full forcing conditions estimated with PlioMIP2 MMM. Right column: $\delta(P-E)$ due to
 vegetation and ice sheet changes estimated with CESM2.

 **Fig. S9** Climatology of preindustrial (shaded) and changes (contour, dashed: negative; solid:
 positive) of a) 850 hPa eddy kinetic energy and b) 200 hPa zonal wind between PliMIP2 and PI
 averaged for CCSM4, CESM1, and CESM2. Hatches show at least one model with insignificant
 changes compared to the 100 model year interannual variability through Student's t-test with
 $p > 0.1$.

 **Fig. S10**
 $\delta(P-E)$ across a) Sahel and b) subtropical east Asia (red boxes in Fig. 4) of individual PlioMIP2
 simulations as a function of changes in tropospheric humidity ($\delta(P-E)_o$), stationary wave
 dynamics ($\delta(P-E)_{v^*}$), zonal mean circulation ($\delta(P-E)_{[V]}$), non-linear combination of tropospheric
 humidity and winds ($\delta(P-E)_{QV}$), and a residual term ($\delta(P-E)_{res}$).

 **Fig. S11** Changes of surface temperature (color shaded), wave number 1 (contour), and
 rotational winds (vectors) of stationary wave in response to F_{CO_2} simulated by CESM2. Notice
 that the stationary wave pattern has little dependency on surface temperature changes, but mainly
 reflects the high-pressure system above Tibet developed during the boreal summer.

Fig. S12 Change in the % of latent heat flux in total surface heat flux (the sum of latent and sensible heat flux) due to Pliocene (a) CO₂ and (b) vegetation and ice sheet changes.

Fig. S13 Time series of global mean net top of the atmosphere radiation imbalance (F_{net}) and surface temperature for the entire simulations of E400, Eo400, and Eo280. To reduce computational cost, Eo400 is initialized from the 1200-year full forcing simulations of the mid-Pliocene (Feng et al., 2020). E400 and Eo280 are both initialized from the preindustrial state. As shown in Fig. S8, influences of initialization or background climate states are minimal for the study regions in the long simulations.

**References**

- 1. Pound, M. J., Tindall, J. & Pickering, S. J. Late Pliocene lakes and soils: a global data set for the analysis of
climate feedbacks in a warmer world. *Climate of the ...* (2014).
- 2. Fauquette, S. *et al.* Climate and biomes in the West Mediterranean area during the Pliocene.
*Palaeogeography, Palaeoclimatology, Palaeoecology* **152**, 15–36 (1999).
- 3. Blavoux, B., Dubar, M. & Daniel, M. Indices isotopiques (13C et 18O) d'un important refroidissement du
climat à la fin du Pliocène (formation lacustre de Puimoisson, Alpes-de-Haute-Provence, France): Isotopic
indices (13C and 18O) of an important cooling at the end of the Pliocene (Puimoisson lacustrine formation,
Alpes-de-Haute-Provence, France). *Comptes Rendus de l'Académie des Sciences-Series IIA-Earth and*
*Planetary Science* **329**, 183–188 (1999).
- 4. van Dam, J. A. Geographic and temporal patterns in the late Neogene (12–3 Ma) aridification of Europe: the
use of small mammals as paleoprecipitation proxies. *Palaeogeography, Palaeoclimatology, Palaeoecology*
**238**, 190–218 (2006).
- 5. deMenocal, P. B., African climate change and faunal evolution during the Pliocene–Pleistocene. *Earth and*
*Planetary Science Letters* **220**, 3–24 (2004).
- 6. Demenocal, P. B. Plio-pleistocene African climate. *Science* **270**, 53–59 (1995).
- 7. Zazzo, A. *et al.* Herbivore paleodiet and paleoenvironmental changes in Chad during the Pliocene using
stable isotope ratios of tooth enamel carbonate. *Paleobiology* **26**, 294–309 (2000).
- 8. Levin, N. E. Environment and climate of early human evolution. *Annu. Rev. Earth Planet. Sci.* **43**, 405–429
(2015).
- 9. Feakins, S. J., Demenocal, P. B. & Eglinton, T. I. Biomarker records of late Neogene changes in northeast
African vegetation. *Geology* **33**, 977–980 (2005).
- 10. Lupien, R. L. *et al.* Vegetation change in the Baringo Basin, East Africa across the onset of Northern
Hemisphere glaciation 3.3–2.6 Ma. *Palaeogeography, Palaeoclimatology, Palaeoecology* 109426 (2019).
- 11. Campisano, C. J. & Feibel, C. S. Depositional environments and stratigraphic summary of the Pliocene
Hadar formation at Hadar, Afar depression, Ethiopia. *The geology of early humans in the Horn of Africa*
**446**, 179–201 (2008).
- 12. Westover, K. S. *et al.* Diatom paleolimnology of late Pliocene Baringo Basin (Kenya) paleolakes.
*Palaeogeography, Palaeoclimatology, Palaeoecology* 109382 (2019).
- 13. Munoz, A., Ojeda, J. & Sanchez-Valverde, B. Sunspot-like and ENSO/NAO-like periodicities in
lacustrinelaminated sediments of the Pliocene Villarroja Basin (La Rioja, Spain). *Journal of*
*Paleolimnology* **27**, 453–463 (2002).
- 14. Heermance, R. V. *et al.* Climatic and tectonic controls on sedimentation and erosion during the Pliocene–
Quaternary in the Qaidam Basin (China). *Bulletin* **125**, 833–856 (2013).
- 15. Rousseau, D.-D., Parra, I., Cour, P. & Clet, M. Continental climatic changes in Normandy (France) between
3.3 and 2.3 Myr BP. *Palaeogeography, Palaeoclimatology, Palaeoecology* **113**, 373–383 (1995).
- 16. Sanyal, P., Bhattacharya, S. K., Kumar, R., Ghosh, S. K. & Sangode, S. J. Mio–Pliocene monsoonal record
from Himalayan foreland basin (Indian Siwalik) and its relation to vegetational change. *Palaeogeography,*
*Palaeoclimatology, Palaeoecology* **205**, 23–41 (2004).
- 17. Gaur, R. & Chopra, S. Taphonomy, fauna, environment and ecology of upper Sivaliks (Plio-Pleistocene)
near Chandigarh, India. *Nature* **308**, 353–355 (1984).
- 18. IGARASHI, Y., YOSHIDA, M. & TABATA, H. History of vegetation and climate in the Kathmandu
Valley. *Proceedings of the Indian National Science Academy, Part A. Physical sciences* **54**, 550–563 (1988).
- 19. Xie, S. *et al.* Palaeoclimatic estimates for the late Pliocene based on leaf physiognomy from western
Yunnan, China. *Turkish Journal of Earth Sciences* **21**, 251–261 (2012).
- 20. Wang, J., Wang, Y. J., Liu, Z. C., Li, J. Q. & Xi, P. Cenozoic environmental evolution of the Qaidam Basin
and its implications for the uplift of the Tibetan Plateau and the drying of central Asia. *Palaeogeography,*
*Palaeoclimatology, Palaeoecology* **152**, 37–47 (1999).
- 21. Koutsodendris, A. *et al.* Late Pliocene vegetation turnover on the NE Tibetan Plateau (Central Asia)
triggered by early Northern Hemisphere glaciation. *Global and Planetary Change* **180**, 117–125 (2019).
- 22. Yao, Y.-F. *et al.* Monsoon versus uplift in southwestern China—late Pliocene climate in Yuanmou Basin,
Yunnan. *PLoS One* **7**, e37760 (2012).
- 23. Chang, Z., Xiao, J., Lü, L. & Yao, H. Abrupt shifts in the Indian monsoon during the Pliocene marked by
high-resolution terrestrial records from the Yuanmou Basin in southwest China. *Journal of Asian Earth*
*Sciences* **37**, 166–175 (2010).

Commented [FR2]: corrected according to reviewer 1

- 24. Ji, S., Nie, J., Breecker, D. O., Luo, Z. & Song, Y. Intensified aridity in northern China during the middle
Piacenzian warm period. *Journal of Asian Earth Sciences* **147**, 222–225 (2017).
- 25. Sun, Y., An, Z., Clemens, S. C., Bloemendal, J. & Vandenberghe, J. Seven million years of wind and
precipitation variability on the Chinese Loess Plateau. *Earth and Planetary Science Letters* **297**, 525–535
(2010).
- 26. Lu, H., Wang, X. & Li, L. Aeolian sediment evidence that global cooling has driven late Cenozoic stepwise
aridification in central Asia. *Geological Society, London, Special Publications* **342**, 29–44 (2010).
- 27. Gent, P. R. *et al.* The community climate system model version 4. *J. Climate* **24**, 4973–4991 (2011).
- 28. Dowsett, H. *et al.* The PRISM4 (mid-Piacenzian) paleoenvironmental reconstruction. *Climate of the Past*
**12**, 1519–1538 (2016).
- 29. Hurrell, J. W. *et al.* The community earth system model: a framework for collaborative research. *Bull. Amer.*
*Meteor. Soc.* **94**, 1339–1360 (2013).
- 30. Danabasoglu, G. *et al.* The Community Earth System Model version 2 (CESM2). *Journal of Advances in*
*Modeling Earth Systems* **12**, e2019MS001916 (2020).
- 31. Jungclaus, J. H. *et al.* Ocean circulation and tropical variability in the coupled model ECHAM5/MPI-OM. *J.*
*Climate* **19**, 3952–3972 (2006).
- 32. Hazeleger, W. *et al.* EC-Earth V2. 2: description and validation of a new seamless earth system prediction
model. *Clim Dyn* **39**, 2611–2629 (2012).
- 33. Gordon, C. *et al.* The simulation of SST, sea ice extents and ocean heat transports in a version of the Hadley
Centre coupled model without flux adjustments. *Clim Dyn* **16**, 147–168 (2000).
- 34. Marti, O. *et al.* Key features of the IPSL ocean atmosphere model and its sensitivity to atmospheric
resolution. *Clim Dyn* **34**, 1–26 (2010).
- 35. Boucher, O. *et al.* Presentation and evaluation of the IPSL-CM6A-LR climate model. *Journal of Advances*
*in Modeling Earth Systems* **12**, e2019MS002010 (2020).
- 36. Hasumi, H. & Emori, S. K-1 coupled model (MIROC) description. K-1 Technical Report 1. *Center for*
*Climate System Research, University of Tokyo* (2004).
- 37. Zhang, Z. S. *et al.* Pre-industrial and mid-Pliocene simulations with NorESM-L. *Geoscientific Model*
*Development* **5**, 523–533 (2012).
- 38. Hewitt, H. T. *et al.* Design and implementation of the infrastructure of HadGEM3: The next-generation Met
Office climate modelling system. *Geoscientific Model Development* **4**, 223–253 (2011).
- 39. Kelley, M. *et al.* GISS-E2. 1: Configurations and climatology. *Journal of Advances in Modeling Earth*
*Systems* **12**, e2019MS002025 (2020).

REVIEWERS' COMMENTS

Reviewer #1 (Remarks to the Author):

The authors have done a great job of revising the manuscript, and all of my comments have been addressed.